# Medieval and early modern diets in the Polack region of Belarus: A stable isotope perspective

**Vera Haponava**[1]*, **Aliaksei Kots**[2], **Mary Lucas**[3], **Max Both**[4], **Patrick Roberts**[3,5,6]*

**1** Faculty of Archaeology, University of Warsaw, Warsaw, Poland, **2** Department of History and Tourism, Polotsk State University, Novopolotsk, Belarus, **3** Department of Archaeology, Max Planck Institute for Geoanthropology, Jena, Germany, **4** Institute of Prehistoric Archaeology, Freie Universitaet, Berlin, Germany, **5** isoTROPIC Independent Research Group, Max Planck Institute for Geoanthropology, Jena, Germany, **6** School of Social Sciences, University of Queensland, St Lucia, Brisbane, Australia

* harti@protonmail.com (VH); roberts@shh.mpg.de (PR)

**Data Availability Statement:** All relevant data are within the manuscript and its Supporting Information files.

**Funding:** VH, ML, and PR thank the Max Planck Society (https://www.shh.mpg.de/en) for funding

## Abstract

In western and north-western Europe there has been a growing focus on exploring how major economic, political, and social changes during the Medieval period impacted the lived experience of different populations and sectors of society. Stable isotope analysis has proven particularly powerful in this regard, providing direct insights into the long-term diets of individuals and communities. Despite experiencing similarly dramatic social reconfigurations and changes, eastern Europe has, however, received far less attention in this regard. The territory of Belarus has, especially, so far remained a relative blank spot on the bioarchaeological map of Europe, though cities such as Polack emerged rapidly as key nodes within a growing economic and religious network. To gain direct insight into the diets of inhabitants of the Polack region of Belarus in the 11-18th centuries, we applied stable carbon and nitrogen isotope analysis to bone and dentine collagen from human (n = 143) and animal (n = 105) individuals from the city of Polack and surrounding rural sites. Results indicate a diet based on $C_3$ terrestrial resources, which did not differ between sexes and showed limited variation over time. Contrary to expectations, it appears that animal products were commonly consumed by rural dwellers, but no significant reliance on fish resources or millet consumption is found. In contrast to examples from western Europe, we argue that the diets in the city and the surrounding villages remained broadly similar for the majority of the population, and similar to commoners analysed in Poland and Lithuania, perhaps suggestive of slightly different economic changes operating in this part of the Medieval world.

## 1. Introduction

The Medieval period of Europe witnessed a series of major economic, political, and social changes, including the rise of Christian states, the expansion of urban networks together with facilitated disease transmission, the intensification of hierarchies and wealth inequality, and increased access to non-local resources [1–4]. While these broader processes are well-documented historically and archaeologically, their impacts on local populations in different parts

the laboratory work and stable isotope analysis. VH received an additional funding support from the Faculty of Archaeology of the University of Warsaw (https://www.archeologia.uw.edu.pl/en/new-main-page/) in the amount of 2801 PLN towards the cost of trip to Germany for the processing of samples. The funders had no role in study design, data collection and analysis, decision to publish, or preparation of the manuscript.

**Competing interests:** The authors have declared that no competing interests exist.

of Europe are often more obscure, particularly given that archival records are often biased towards elite communities [1,4,5]. In particular, eastern Europe, located beyond the geographic scope of Roman Christianisation, has also received less attention and is often considered to be poorer, less developed, and less "European" [5]. This is despite the fact that eastern Europe also went through dramatic social changes, with the territory of Belarus in particular witnessing the appearance of its first clear cities and states at the end of the first millennium and the beginning of the second millennium AD, the rise and decline of feudal relationships between the 9[th] and the beginning of the 19[th] centuries AD, a number of major military conflicts and constant raids, the formation of land relations (e.g. adoption of the the so-called Voloka Decree), and the appearance of Magdeburg Law in the cities [6–10]. Being part of the Grand Duchy of Lithuania, at its height the largest state in Medieval Europe, and described in the literature as a melting pot of ethnicities, religions and cultures [11], the region thus remains a yet untapped opportunity to explore issues pertaining to identity, equality, human agency, resilience and adaptation to socio-political transitions in the Medieval world through the use of the state-of-the-art methods offered by bioarchaeology.

The study of individual and group diets offers the possibility of exploring the interaction of cultural, political, and economic change in the lived experience of Medieval populations. Various methodologies can be applied to study past human diets. However, it is in combination where they provide the most comprehensive information. While yielding high levels of detail on cuisine and foodstuffs consumed, written records are still virtually nonexistent for the first centuries of the 1[st] millennium AD in certain parts of eastern Europe [9,12,13] and, where available, historical documentation revolves around social and religious elite groups [5]. Zooarchaeological and archaeobotanical data speak to the specific foodstuffs available for a given, context-based window of time but do not necessarily provide insights into the long-term dietary reliance of an individual [12]. They are also limited by taphonomic biases [14] and retrieval techniques, and are not always capable of showing how available resources were distributed within the population through time [3]. Stable isotope analysis of human remains has emerged as a means of obtaining direct and quantitative information about the diet of individuals and communities [12,14,15]. It can also, when combined with other more traditional methodologies, enable the exploration of the diet of individuals overlooked in the Medieval historical record, such as juveniles, individuals from lower social statuses, and those residing in nonurban locations [5].

Expanding isotopic studies in western and north-western Europe have demonstrated the role of religious prescription, status, and political upheaval in shaping diets and economies [3,4,16,17]. However, this technique has been rarely applied in an eastern European Medieval context [5], despite its potential to answer similar, albeit more region-specific, questions. In this study we apply stable carbon and nitrogen isotope analysis to a large sample of individuals from 15 rural graveyards and the inhabitants of one major city from the northern part of Belarus, as well as animal remains retrieved from several contemporaneous sites in the region. The time span of the sites covers the period from the 11[th] till the 18[th] centuries AD, and is subdivided into three roughly equal periods corresponding to periods of major state formation within the territory—Ancient Rus principalities (9-12[th] century AD), the Grand Duchy of Lithuania (13-16[th] century AD) and the Polish-Lithuanian Commonwealth (16-18[th] century AD). We refer to them as High, Late Medieval and Early Modern periods following the general division accepted in European history here, though we admit the local variation of this periodisation. Our goal is to determine whether there were temporal dietary changes brought by the associated political, economic, and social transitions, both in urban and rural settings, in male and female individuals, and between the groups defined by archaeologists as nobility and military based on their burial context versus the rural individuals. We note that this division is

fairly simplistic and may not always accurately reflect lived experience or status but, nevertheless, we believe that some broad insights into social and political elements of diet in past Belarus can be developed in this regard. We also seek to compare our data to that available from the neighbouring regions of Poland and Lithuania to explore how diets varied across the expanding states of the region.

## 2. Background

### 2.1 Diet in 11-18th-century Belarus based on historical, archaeological and ethnographic sources

The second millennium AD witnessed dramatic social, political, and cultural changes within the territory of the current nation of Belarus, including the appearance of a clear, established urban network. Formed on the trans-European trade route "from the Varangians to the Greeks", described in the Russian Primary Chronicle as connecting Scandinavia (Varangians) with Byzantium [18,19], the city of Polack (also Polatsk, Polotsk; official transliteration from the Belarusian language is used here for Belarusian toponyms and names, spelled with Belarusian Latin alphabet) was the major trade, political and cultural centre of the region [7,9]. The Polack Principality was the most prominent of the proto-states on the Belarusian lands, having its own ruling dynasty and maintaining certain independence even when it became part of larger territorial formations in this part of Eurasia [9,20]. Over the course of the period between the 11th and 18th centuries, these formations included Ancient Rus, the Grand Duchy of Lithuania (GDL), and the Polish-Lithuanian Commonwealth (PLC). Being one of the multiple states that together comprised Ancient Rus, the Polack Principality was involved both in prospering trade and numerous military campaigns and territorial changes [13,21]. Against the backdrop of battles with Tatar Mongols and the Teutonic Order a new state formed in Eastern Europe in the middle of the 13th century—the Grand Duchy of Lithuania, and from the 14th century AD, the lands of Polack became part of it. In the 15-16th centuries the culture of GDL was impacted by the European Renaissance; cities, craft and education flourished [22]. In 1569 the GDL and Poland united into the Polish-Lithuanian Commonwealth, and the following centuries were marked by devastating wars, depopulation and decline of the role of Polack as a political and socio-economic centre [9]. The PLC ceased to exist at the end of the 18th century when it was partitioned by Austria, Russia and Prussia, and the Polack region, as well as other Belarusian lands, was gradually annexed by the Russian Empire (refer to S1 Note in S1 File for more details on the historical background). As social identity is believed to be deeply embedded in diet and consumption behaviours [3], these events might have been paralleled by dietary changes. So far the studies related to nutrition in Belarus have focused on written sources [23,24], ethnohistoric and archival data [25–27] and archaeological materials such as animal remains [28] and charred grain finds [29,30]. These shed light on the major food products and the ways of their preparation through the discussion of agriculture, husbandry and hunting, trade, or the menu of nobility.

Based on these studies we may assume that throughout the 11-18th centuries a plant-based diet dominated across the territory of Belarus, with grains constituting the base of the local inhabitants' subsistence [10,25,31]. Rye bread was the main food product for centuries, as well as gruel [7,27,32]. Rye, followed by wheat, barley and oats were the most popular cereals [10,29,30,33], while garden crops like cabbage and turnip were also common [7,25,33–35]. By contrast, millets appear in rather negligible quantities among charred grain finds and continue to decrease in number over the period of our focus [29], though some authors name millets as a widely grown crop and even one of the main foods in Ancient Rus [7,33]. As millet was found to be underrepresented in charred grain assemblages [36], this may also be due to

taphonomic reasons. In such a light, isotope analysis can provide a means to directly validate the extent of consumption of millets. It is similarly hard to estimate the role of legumes, but overall they seem to appear in rather small quantities as well [29,30]. Manuring is considered a necessary component of the crop rotation system, which was the dominant agricultural system throughout the period of our study [7,10,30,33,35]. However, based on written and archaeological sources there is a possibility that manuring was not used or had limited application in northern Belarusian territories between the 12-18[th] centuries [10,37].

The role of husbandry in the economy was apparently secondary throughout the 11-18[th] centuries, though it still provided vital subsistence components like meat and milk products [7,25,33]. Fat, milk, butter or oil were common additives to non-meat dishes like porridge, bread, and cabbage, and peasants have been argued to have eaten meat only rarely, mostly during festive dinners [26,32]. Pig was the dominant domestic animal, followed by cattle [28,38]. Hunting, fishing, bortfreat (collection of wild honey), and gathering of mushrooms, berries, nuts and medicinal herbs occupied an important place in the ancient peasant economy, providing additional sources of food [7,25,33]. At the same time, the role of hunting was marginal [28]—only for the feudal aristocracy did hunting remain an important social activity [7]. The territory surrounding Polack, Paazerje, is rich in water resources [39], and fishing was a widespread occupation [7,40]. Commonly caught species included perch, bream, tench, roach, ide, pike, catfish, sea and river lamprey, trout, carp, eel, and pike perch (zander) [7,39,41]. Christian fasts lasted, in sum, almost half a year [5,25,42], thus a substantial influence of the Christian religion on fish consumption may be expected for the whole period of our study. Fast was reportedly kept very strictly, even by breastfeeding women and by children starting from 7 years of age [32].

Based on the available information it appears that commoners living in towns would have had diets similar to those of rural commoners. Despite agricultural activity and husbandry being common in cities including Polack in the 10-16[th] centuries, city inhabitants were the main consumers of village production, increasingly obtaining their food from rural neighbourhoods [7,9,29,30,33,37,41]. The same was likely true for the later period of the 16-18[th] centuries [43]. One possible difference between the diets of rural and urban dwellers could be that foreign goods, including food products, almost never reached rural areas [7], while Polack citizens already obtained some imported foods and drinks from the very beginning of the period of our focus [9]. However, any imports of foodstuffs seem to have focused on rather luxurious products, such as wine, oil, salt, spices, fruits, and salted herring [7,9,44–46]. Therefore, if differences based on access to rare foreign foods existed between rural and urban dwellers, they were likely rather negligible and more applicable to wealthy citizens than commoners. The diet of the elites from the times of Ancient Rus has been argued to be more abundant overall and richer in meat in particular [47]. On the other hand, Navahrodski [48] proposes that, until the 16[th] century, social elites in Belarusian lands consumed almost the same products as the commoners, maybe only of better quality and more regularly. From the 16[th] century, the cuisine of the nobility was influenced by Poland, for which excessive fat content, presence of sweet dishes, a lot of spices and salt, and consumption in great quantities were typical, testifying the high status of an individual. Additionally, with time, nobility and rich citizens started to use more and more imported products [48] like citrus fruits, potatoes (later), foreign fish, caviar, and even snails. Wine was also imported in large quantities [23] (refer to S2 Note in S1 File for additional information on past diets in Belarus). Overall, historical and archaeological evidence indicates that diet variability and access to nonlocal resources increased throughout Europe during the Medieval and Early Modern periods [5]. Stable carbon and nitrogen isotope analysis may shed light on some aspects of the diet described above, such as the dominance of plant-based diets and the differing access to animal protein among various groups like elites,

rural and urban dwellers. It has also proven useful in detecting the consumption of millets and fish, while potential enrichment in $^{15}$N of domestic species may support the use of manure on crops that they would then subsequently feed. Other changes in diet, such as increased incorporation of imported foods may be only observed if these foods were isotopically distinguishable—for example, marine products.

## 2.2. Stable isotope analysis and palaeodiet in Medieval eastern Europe

The use of stable isotope data in palaeodietary studies is based on the fact that stable isotopic ratios of carbon and nitrogen in the body reflect those of the food consumed during life [3,15]. Stable carbon isotope values of bone collagen (expressed as $\delta^{13}$C (‰)) have traditionally been used to distinguish between dietary protein originating from plants (and animals consuming these plants) utilising the $C_4$ photosynthetic pathway (e.g. maize and millet) versus $C_3$ plants (most temperate species, e.g. wheat and barley). It is also used to distinguish between marine and terrestrial protein consumption, with marine environments having a different source of $CO_2$ that leads to an overlap with $C_4$ resources [3,12,14,15,42]. In Europe, there are only rare examples of wild $C_4$ plants, primarily found in the Mediterranean, plus domestic species which have been introduced at various points, the most important of which are the millets prior to European contact with the Americas [14]. In the case of bone collagen, it is worth noting that the $\delta^{13}$C signal will be primarily reflective of protein contributions to the diet. This means that high protein (e.g. meat, aquatic) resources can swamp the signal at the expense of lower protein (e.g. crop) resources [49]. There is an approximate 1‰ increase in $\delta^{13}$C with each trophic level [50].

Stable nitrogen isotope analysis of bone collagen ($\delta^{15}$N) provides additional resolution of the trophic position of an individual within the food chain. In archaeological contexts, this includes testing the proportional consumption of animal (meat or dairy) versus plant protein or aquatic versus terrestrial protein, as both freshwater and marine ecosystems tend to, on average, have longer food chains resulting in higher $\delta^{15}$N values (although freshwater settings have much more variable $\delta^{13}$C than marine ecosystems) [3,12,14,15,42]. Not all variation in stable isotope ratios is directly attributable to differences in foods consumed among individuals or populations, however [3]. Other factors that may affect the $\delta^{13}$C and $\delta^{15}$N values of food sources for agricultural human populations and their domesticates include physiology (starvation, nursing), environmental factors such as aridity, farming practices such as manuring which leads to $^{15}$N enrichment of soils and thus plants, the "canopy effect" which leads to lower $\delta^{13}$C values in wild animals occupying closed forest environments compared to domesticated animals occupying open areas, and variation in herding and grazing practices for different livestock taxa [12,42].

Although there is currently a complete absence of isotope-based studies dedicated to palaeodiet on the territory of Belarus, we can turn to those conducted in the neighbouring regions as a reference. For example, a series of isotope studies published by Reitsema, Kozłowski and others [2,3] investigated palaeodiet in Poland for the period ranging from Medieval Poland of the 10-14th centuries to the Polish-Lithuanian Commonwealth of the 16-18th centuries. These studies found dramatic "status-based" differences in diet in the Commonwealth, with much fish consumed by elites, and detected certain trends like the lack of consistent diet changes over time and presence of site-specific changes. In Lithuania, research by Skipitytė et al. [42] and Whitmore et al. [5] showed a certain contradiction to the historical sources and unexpected results in terms of little evidence for fish consumption. Stable isotopes were also used for the examination of palaeodiet in the 13th-century city of Yaroslavl, Russia, where a substantial proportion of animal protein in the diet was observed [51].

## 3. Materials and methods

### 3.1. Sites and burials

The human remains analysed in this study originate from burial sites that include kurgan (burial mound) and ground (pit burial) cemeteries in rural and urban environments. Early ground cemeteries partly contemporaneous to kurgans are characterised by stone constructions in the form of a pavement or lining, often with larger stones at the head and/or foot of the burial [52–54] (refer to S1 Fig in S1 File, Supplementary Information, for an example of a kurgan burial and S2 and S3 Figs in S1 File for examples of ground burials). Such 'stone graves' are mostly dated to the 13-14th centuries AD, though they existed until the 18th century [53]. Kurgans and flat graves are frequently combined in long-used burial complexes throughout the region. A large concentration of such burial complexes is found in the south of the Belarusian Dzvina (also Dvina, Western Dvina, or Daugava) basin—the territory of our study. During most of the period of our focus, in the 14-18th centuries AD, the deceased were interred without any grave goods [52,54]. Rural cemeteries that provide the sample of village population in this study include Biruli, Domžarycy, Doŭhaje, Dubraŭka, Dzmitraŭščyna, Ivieś, Kazloŭcy, Klieščyno, Michalinava, Padsvillie, Pieravoz-4, Sielišča, Skrabianiec, Vaŭča and Ziabki. Fig 1 shows a map of the sampled sites grouped into the three periods (11-13th, 13-16th, 16-18th centuries AD) of major state formation mentioned above (Ancient Rus, the GDL, the PLC).

The dating and cemetery types of each site along with the number of individuals excavated and sampled from them can be found in Table 1. In several sites in this study, the sampled material comes from two phases. Namely, these are the kurgan and later ground burials of Biruli (11-12th centuries AD and the 14th—beginning of the 16th century AD) [55,56], the site of

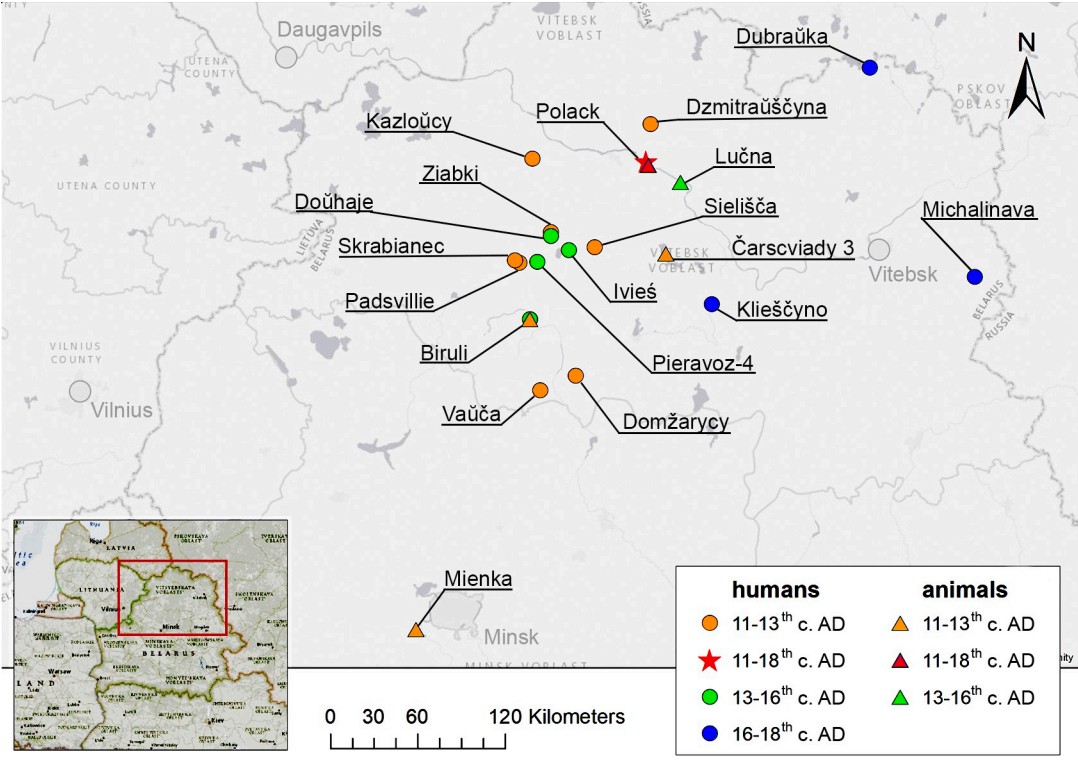

**Fig 1. Map of the sites from which human and animal materials were sampled, colour-coded by their dating.** The red star shows the urban centre of Polack while all other sites are designated as 'rural'.

Table 1. Summary of cemetery sites from which human material originated, including cemetery type (K stands for "kurgan" and G stands for "ground"), site type (urban/rural), the context of the burial (elite, military, non-elite), dating, number of individuals excavated and number of individuals sampled.

| Site name | Cemetery type | Site type | Context | Dating, centuries AD | # ind. excavated | # ind. sampled |
|---|---|---|---|---|---|---|
| **Biruli** | K | rural | non-elite | 11–12 | 20 [60] | 1 |
| | G | rural | non-elite | 14–16 | 13 [56,61] | 9 |
| **Domžarycy** | K | rural | non-elite | 11 | 12 [62] | 4 |
| **Doǔhaje** | G | rural | non-elite | 14–16 | 31 [63,64] | 5 |
| **Dubraǔka** | G | rural | non-elite | 17–18 | 24 [65] | 6 |
| **Dzmitraǔščyna** | K | rural | non-elite | 11 | 3 [66] | 1 |
| **Ivieś** | K | rural | non-elite | 12 | 1 [67] | 1 |
| | G | rural | non-elite | 14–16 | 28 [52,67–69] | 10 |
| **Kazloǔcy** | K? | rural | non-elite | 10–12 | 7 [70] | 2 |
| **Klieščyno** | G | rural | non-elite | 16–18 | 9 [71] | 3 |
| **Michalinava** | G | rural | non-elite | 18–19 | 30 [72] | 11 |
| **Padsvillie** | K | rural | non-elite | 12 | 2 [73,74] | 2 |
| **Pieravoz-4** | K | rural | non-elite | 11–12 | 12 [74,75] | 3 |
| | G | rural | non-elite | 13–14 | 17 [57,59,75] | 7 |
| **Sielišča** | K | rural | non-elite | 10–12 | 10 [76] | 2 |
| **Skrabianiec** | K | rural | non-elite | 11 | 7 [73] | 3 |
| **Vaǔča** | K | rural | non-elite | 12–14 | 6 [58] | 2 |
| | G | rural | non-elite | 13–14 | 20 [58] | 7 |
| **Ziabki** | K | rural | non-elite | 11–12 | 7 [77] | 2 |
| **Polack—Lower Castle** | G | urban | non-elite | 13–14 | 35 [78] | 18 |
| | G | urban | military | 18 | 11 [78] | 10 |
| **Polack—Township** | G | urban | elite | 17–18 | 48 [79,80] | 13 |
| **Polack—Upper Castle** | G | urban | non-elite | 11–13 | 115 | 21 |
| | | | | Total from the period of the **11-13th** centuries: | | 44 |
| | | | | Total from the period of the **13-16th** centuries: | | 56 |
| | | | | Total from the period of the **16-18th** centuries: | | 43 |

Pieravoz (kurgan burials are dated to the 11th—beginning of the 12th century AD, while the ground part of cemetery is dated to the 13-14th centuries AD; refer to the situational plan of this site in S4 Fig in S1 File, Supplementary Information) [57] and Vaǔča (kurgans dated to the 12-14th centuries AD, while 'stone graves' date to the 13-14th centuries AD) [58,59]. In fact, Doǔhaje (a ground cemetery of the 14-16th centuries AD) and Ziabki (kurgans of the 10-12th centuries AD) are located so close that their burials represent another long-functioning common complex. Thus, the rural sample covers the whole timespan of focus of this study, with the groups from each period being roughly similar in number (see Table 1 for the dating, provenance, and number of individuals sampled and S3 Note in S1 File for further information about the sites).

The city of Polack played the role of the administrative, social and economic centre of the region under study in the 11-16th centuries AD [39]. The human and animal material comes from several parts of this city—the Lower Castle, Upper Castle, and Township. The skeletal material from these sites represents the urban sample in this study and, like the rural sample, quite evenly covers the whole timespan of the 11-18th centuries (Table 1).

A multi-layered graveyard was excavated in the territory of the Lower Castle, yielding individuals from burials near an Orthodox church dated to the 13-14th centuries AD and individuals dated to the 17th century AD—mostly young males interpreted as having military affiliations based on finds of bullets with some of the deceased [78]. The Township was the

**Table 2. Summary of sites from which animal material originated.**

| Site name | Dating, centuries AD | Species | # individuals sampled |
|---|---|---|---|
| Biruli | 10–11 | beaver, cattle, horse, pig/boar, sheep/goat | 12 |
| | 11–13 | sheep | 1 |
| Čarscviady 3 | 11–12 | bison?, cattle, chicken, fish, hare, pig, sheep/goat, wild bird | 13 |
| Lučna | 15–16 | cat, cattle, goose, rabbit, pig, sheep/goat | 14 |
| | 16–17 | cattle, horse/pony | 2 |
| Mienka | 10–11 | carnivore (dog?), cattle, dog/wolf, elk, horse, pig, sheep/goat | 23 |
| Polack—Lower Castle | 10–11 | beaver, cattle, chicken, galliformes, pig/boar, pike, sheep | 13 |
| | 11–13 | cattle, sheep | 3 |
| | 13-14(?) | cattle, goat, horse, pig | 5 |
| | 14–15 | cattle, rabbit, sheep/goat | 3 |
| | 17–18 | chicken, galliformes, sheep/goat | 3 |
| | 18–19 | cattle, pig, red deer | 5 |
| Polack—Upper Castle | 16 | cattle, elk, horse | 3 |
| | 17 | horse | 1 |
| | 18 | horse | 1 |
| | 17–18 | pig | 1 |
| | 18–19 | cattle, pig | 2 |
| | | Total from the period of the 11-13th centuries: | 65 |
| | | Total from the period of the 13-16th centuries: | 25 |
| | | Total from the period of the 16-18th centuries: | 15 |

ancient centre of Polack, which was turned into a cemetery in the 17-18th centuries AD. This cemetery was excavated in 2007 and 2009 and is interpreted as being Catholic from the scarce inventory of burial goods uncovered. Based on the Catholic confession, and on the fact that the cemetery is located in the centre of the city, the researchers concluded that the deceased belonged to Polack nobility [79]—thus this group constitutes the elite sub-sample which will be compared to rural individuals in our study. The ground necropolis on the territory of the Upper Castle was discovered in 2020, and at the time of this research 115 burials dated to the 11-13th centuries AD had been excavated by I.U. Mahalinski and A.L. Kots (S3 Note in S1 File).

### 3.2 Faunal remains

Due to the fact that the exact isotopic ratios at the base of the food chain can vary across space and time as a product of various environmental and economic parameters it is necessary to sample contemporaneous fauna to fully understand and interpret the human results [15]. Part of the animal sample in this study originates from the same sites as human remains—the city of Polack (Lower Castle and Upper Castle) and the settlement site and one of the kurgans of Biruli. The rest of the animal remains come from other sites associated with the Polack Principality territory—Lučna (15-17th centuries AD) and Čarscviady (11-12th centuries AD), rural sites that hosted nobility residences, and the Mienka township (10-11th centuries AD) (Table 2). Fauna was identified by Dr. Urszula Iwaszczuk and Vera Haponava using the reference collection at the Institute of Mediterranean and Oriental Cultures of the Polish Academy of Sciences and a standard zoological reference work [81,82].

### 3.3 Samples

A total of 100 human bones and 43 human teeth were sampled in this study, representing 5–100%, on average 44%, of the (known) excavated individuals at each site (refer to Table 1 for

the numbers and to S3 Note in S1 File for the explanation of how the numbers were counted). The individuals and the elements sampled can be seen in S1 Table in S1 File (Supplementary Information). Human bone collagen was sampled primarily from rib bones (92/100), and, in several cases where ribs were lacking, from femur, fibula, ulna, and radius bones. Where post-cranial elements were not available in the collection, permanent maxillary or mandibular third or second molars were preferentially sampled for dentine collagen to reflect the adolescent diet of the individuals. When pathological conditions were expressed on the bones or teeth (e.g. carious lesions), non-affected elements were prioritised for sampling. Given the heterogeneity of samples, the results should be interpreted with caution, as different periods of life are compared here: ribs represent approximately last five years of life [83], while dentine from the roots that were cut for analysis in this study represent the ages of 14–20 in the case of the third molars and 8–14 in the case of second molars [84].

For faunal remains, bones (e.g. ribs) (80/105) or teeth developing later in life (such as M3, M2 or premolars in some species) were preferentially selected for collagen extraction (see S2 Table in S1 File, Supplementary Information). To ensure that samples did not come from the same individual, bones and teeth of the same species from different archaeological contexts, unique elements of the same species from the same archaeological context, or elements clearly belonging to different individuals (e.g. of different age-at-death) were used. The permits for the study and analysis of samples were obtained for the described study from the institutions holding them in collections (Polotsk State University and the Institute of History of the National Academy of Sciences of Belarus).

Sex estimation of human individuals was based on pelvic and cranial morphology [85]. Only adult individuals were included in the study with the exception of 12 adolescents (i.e. only individuals with the second molar fully formed and erupted). The age-at-death was assessed observing cranial suture closure following Piontek [86] and epiphyseal fusion for younger individuals based on Herrmann et al. [87]. Biological profiles for humans from the collection of the Polack State University were assessed by V.A. Yemialyanchyk and D.S. Gritskevich.

In the collection of the Institute of History of the National Academy of Sciences, postcranial skeletal elements were not available. Age was estimated using dental wear patterns [88] coupled with analysis of the sphenooccipital synchondrosis fusion [89]. Our decision in favour of the age-at-death estimation method by Brothwell [88] was based on the fact that Brothwell's sample consisted of prehistoric to Medieval English skeletons while the later Lovejoy [90] method is based on a North American population with hunter-gatherer diets. Therefore, the former is more population specific for the Medieval and Early Modern European material we are dealing with in this study. Sex assessment was based on cranial morphology [85]. In the cases where sex determination based on the study of the cranium contradicted that stated in the available publication or archaeological report, the latter assessments were used as the sex could have been estimated with more certainty at the time of excavation when postcranial elements were also available. Given that sex assessment based on cranium is less reliable, we conduct an additional check before interpreting the results of male to female comparisons. Also due to the necessity to rely on the prior anthropological analysis in the case of individuals from Polack State University osteological collection and recognizing the low reliability of the aging method based on cranial sutures, we report the age-at-death only at the level of age categories "adult" and "adolescent" in S1 Table in S1 File and do not undertake any age-related analysis in the study, using the age estimation solely to establish that the individuals in the sample were predominantly adults.

**3.3.1 Stable isotope measurement.** The standard modified-Longin procedure described in Richards and Hedges [91] was used for collagen extraction. First, the surface of the bone

and tooth samples was superficially cleaned through gentle sand ablation. 1 g samples of bone and tooth roots were obtained by use of a diamond cutting blade. Approximately 1 g of cleaned bone or tooth root was then demineralised in 10 mL aliquots of 0.5 M HCl. The acid was changed every 48 hours for 5–17 days until demineralisation was achieved. Then, the demineralised sample was washed three times in Milli-Q® distilled water and gelatinised using pH 3 HCl at 70°C for 48 h. The resulting solution was filtered into a test tube using an EZEE filter and frozen for 24 hours. The frozen sample was then lyophilised in a freeze dryer until fully dried. 1 mg aliquots of dried collagen were weighed into tin capsules for analysis.

Samples were combusted in a Thermo Scientific Flash 2000 Elemental Analyser coupled to a Thermo Delta V Advantage Mass Spectrometer at the Isotope Laboratory, Max Planck Institute for the Science of Human History (Jena, Germany). All isotopic measurements refer to the ratio between heavy and light isotope ($^{13}C/^{12}C$ or $^{15}N/^{14}N$) measured as δ values in parts per mil (‰) calibrated using a two-point calibration between a series of International Standards (IAEA-N-2 Ammonium Sulfate: $δ^{15}N$ = +20.3 ± 0.2‰, USGS40 L-Glutamic Acid: $δ^{13}C$ = −26.389 ± 0.042‰, $δ^{15}N$ = −4.5 ± 0.1‰, IAEA-CH-6 Sucrose: $δ^{13}C$ = −10.449 ± 0.03‰) and in-house laboratory standards (fish gelatin: $δ^{13}C$ = −15.7 ± 0.1‰, $δ^{15}N$ = −14.3 ± 0.1‰; Urea: $δ^{13}C$ = −41.30 ± 0.04‰, $δ^{15}N$ = −0.32 ± 0.2‰; ± 1 standard deviation is reported here). All samples were measured in duplicate. Analytical error was studied through the repeated measurement of an in-house fish gelatin standard (-15.7 ± 0.3‰ for $δ^{13}C$ and 14.3 ± 0.2‰ for $δ^{15}N$; ± 1 std. dev.).

**3.3.2 Statistical analysis.** Statistical analyses were conducted in STATISTICA 14. A Shapiro-Wilk test was performed to test if the human and animal bone collagen $δ^{13}C$ and $δ^{15}N$ data within the analysed groups were normally distributed. Levene's test was performed to check for homogeneity of variances. Parametric data was then analysed using Student's two-sample t-tests for independent samples and one-way ANOVA tests with post-hoc Bonferroni Tests for multiple pairwise comparisons. When the isotopic data were found to follow a non-normal distribution, non-parametric Mann-Whitney U and Kruskal–Wallis tests were employed, with subsequent Kruskal-Wallis multiple comparisons in the latter case. To determine if a linear relationship exists between $δ^{13}C$ and $δ^{15}N$ values of bone collagen linear regression models were used. In all cases, the results were considered significant if the p-value was lower than 0.05. To assess the magnitude of differences in the mean values for the independent samples t-test with varying sample sizes and standard deviations, we calculated the effect size index Hedges' g with the online software Social Science Statistics [92]. Means are reported ± 2 standard deviations in the below text, unless indicated otherwise (e.g. in figures and tables).

# 4. Results

To ensure that samples were sufficiently preserved and free from diagenetic contamination, the atomic carbon to nitrogen ratio (C:N) was used as a primary criterion according to the parameters published by van Klinken [93]. Collagen yields in the whole sample ranged from 0.002 to 26.36%. The animal sample which had a yield below 1% (Bir13-N2.01) was excluded from the analysis even though its C:N ratio was within the acceptable range of 2.9–3.6. Three more animal samples were excluded from the analysis as their C:N ratio was outside of the acceptable range of 2.9–3.6 (Bir13-N5-6.02, Bir14-N1A.01, and PUC16-NV2.3.03). Therefore, the final number of samples used in the analyses was 244 out of 248 (S2 Table in S1 File).

## 4.1 Faunal remains

The descriptive statistics (range, mean, standard deviation and median) for animal groups by feeding category (i.e. herbivore, omnivore), time period, species and sites can be found in

**Table 3. Summary data for δ¹³C (‰ VPDB) and δ¹⁵N (‰ AIR) of animals analysed in this study displayed by consumer type, domestic or wild animal type, species, period groups, groups by sites.** In the latter case due to the low number of domestic herbivore samples from Čarscviady (n = 3) they were considered for the purposes of statistical comparisons in a group with Lučna, as these sites are closely located and similar in nature (rural sites with nobility residences). Domestic herbivore samples from two sites in Polack (Upper and Lower Castle) were also considered together, as the sample from Upper Castle was small (n = 4).

| | n | δ¹³C Range | δ¹³C Mean | δ¹³C 1 SD | δ¹³C Median | δ¹⁵N Range | δ¹⁵N Mean | δ¹⁵N 1 SD | δ¹⁵N Median |
|---|---|---|---|---|---|---|---|---|---|
| **Herbivores (domestic)** | 49 | -25.4 to—20.4 | -22.8 | 1.1 | -23.0 | 4.7 to 10.4 | 6.6 | 1.1 | 6.3 |
| ***Domestic herbivores 11–13*** | 26 | -23.6 to -20.4 | -22.3 | 1.0 | -22.5 | 5.1 to 10.4 | 6.8 | 1.3 | 6.3 |
| ***Domestic herbivores 13–16*** | 17 | -25.4 to -21.2 | -23.3 | 0.9 | -23.1 | 4.7 to 8.7 | 6.3 | 1.0 | 6.3 |
| ***Domestic herbivores 16–18*** | 6 | -24.7 to -22.7 | -23.7 | 0.7 | -23.7 | 5.6 to 7.3 | 6.4 | 0.8 | 6.5 |
| ***Domestic herbivores (rural)*** | 21 | -24.6 to -20.6 | -22.7 | 1.0 | -22.9 | 5.3 to 10.4 | 7.0 | 1.3 | 7.1 |
| *Domestic herbivores (Lučna and Čarscviady)** | 13 | -24.6 to -22.7 | -23.2 | 0.6 | -23.1 | 5.3 to 8.7 | 6.8 | 1.2 | 6.3 |
| *Domestic herbivores (Biruli)* | 8 | -23.6 to -20.6 | -21.9 | 1.1 | -21.7 | 5.6 to 10.4 | 7.4 | 1.5 | 7.3 |
| ***Domestic herbivores (urban)*** | 28 | -25.4 to -20.4 | -22.9 | 1.1 | -23.2 | 4.7 to 9.0 | 6.3 | 0.9 | 6.1 |
| *Domestic herbivores (Polack)* | 19 | -25.4 to -21.2 | -23.3 | 0.9 | -23.4 | 4.7 to 9.0 | 6.3 | 1.1 | 6.5 |
| *Domestic herbivores (Mienka)* | 9 | -23.3 to -20.4 | -22.1 | 1.0 | -22.3 | 5.5 to 6.7 | 6.1 | 0.4 | 6.1 |
| Cattle | 28 | -23.6 to -20.4 | -22.4 | 1.0 | -22.7 | 5.1 to 8.4 | 6.5 | 0.9 | 6.3 |
| Sheep/goat | 12 | -23.8 to -21.5 | -23.0 | 0.6 | -23.2 | 5.2 to 10.4 | 7.4 | 1.5 | 7.2 |
| Horse/pony | 7 | -24.7 to -21.2 | -23.4 | 1.1 | -23.7 | 4.8 to 7.1 | 6.1 | 0.8 | 6.1 |
| Pig/boar | 27 | -25.3 to -21.4 | -23.0 | 0.8 | -23.0 | 4.5 to 10.7 | 7.7 | 1.4 | 7.4 |
| Chicken | 5 | -23.9 to -22.4 | -23.2 | 0.6 | -23.4 | 7.7 to 11.3 | 9.4 | 1.3 | 9.1 |
| **Herbivores (wild)** | 8 | -24.8 to -22.4 | -23.4 | 1.0 | -23.0 | 2.8 to 6.9 | 4.5 | 1.3 | 4.2 |
| **Carnivores** | 4 | -23.9 to -21.6 | -22.8 | 1.0 | -22.8 | 8.7 to 9.3 | 9.0 | 0.3 | 9.0 |
| **Birds** | 10 | -26.0 to -22.4 | -23.6 | 1.0 | -23.4 | 6.5 to 11.3 | 8.4 | 1.5 | 8.4 |
| **Fish** | 3 | -28.2 to -27.6 | -27.8 | 0.3 | -27.8 | 10.9 to 13.5 | 12 | 1.3 | 11.6 |

Table 3, while the plot of the animal δ¹³C and δ¹⁵N values is shown in Fig 2. Wild species in our sample included beaver, hare, bison, elk, red deer and wild bird. All excluding the unidentified wild bird are herbivorous animals (n = 8). Birds (n = 10) are mostly represented by chicken, but also geese and birds only defined to the level of 'galliformes' or wild bird. The fish (two pikes and one unidentified species) bone samples provide approximate δ¹³C and δ¹⁵N for available freshwater resources. Freshwater, anadromous and marine species from studies conducted in Poland and Lithuania serve to complete the picture of potential food sources in this study.

When faunal δ¹³C and δ¹⁵N across sites are compared, only differences in δ¹³C among domestic herbivores appear to be significant based on one-way ANOVA (F = 7.102; df = 3; p < 0.001). However, the differences appear to be 'site-based' and do not simply fall along a rural and urban division. In particular, post-hoc Bonferroni tests (S3 Table in S1 File) show that the Polack sites and rural sites with nobility residences Lučna and Čarscviady exhibit lower δ¹³C values than the rural site Biruli and the township Mienka (Table 3, S6 Fig in S1 File). At the same time, differences in the domestic herbivore δ¹⁵N values, not significant when compared with Kruskal–Wallis test between the groups of sites mentioned above (H = 6.052; df = 3; p = 0.109; S4 Table in S1 File), are significant at the level of the rural to urban sites comparison based on a two-sample t-test (t = 2.342; df = 33.568; p = 0.025). The δ¹⁵N of rural domestic herbivores is on average 0.8‰ higher than that of urban fauna; the effect size of this difference in nitrogen isotope values was moderate to large (Hedges' g = 0.7). Comparing the set of domestic herbivore samples by period with a Kruskal–Wallis test shows that there are differences (H = 15.319; df = 2; p = 0.001) in δ¹³C values between the 11-13th centuries (n = 26) and both the 13-16th centuries (n = 17) and the 16-18th centuries (n = 6), with the mean of the High Medieval fauna being 1‰ higher than that of the Late Medieval

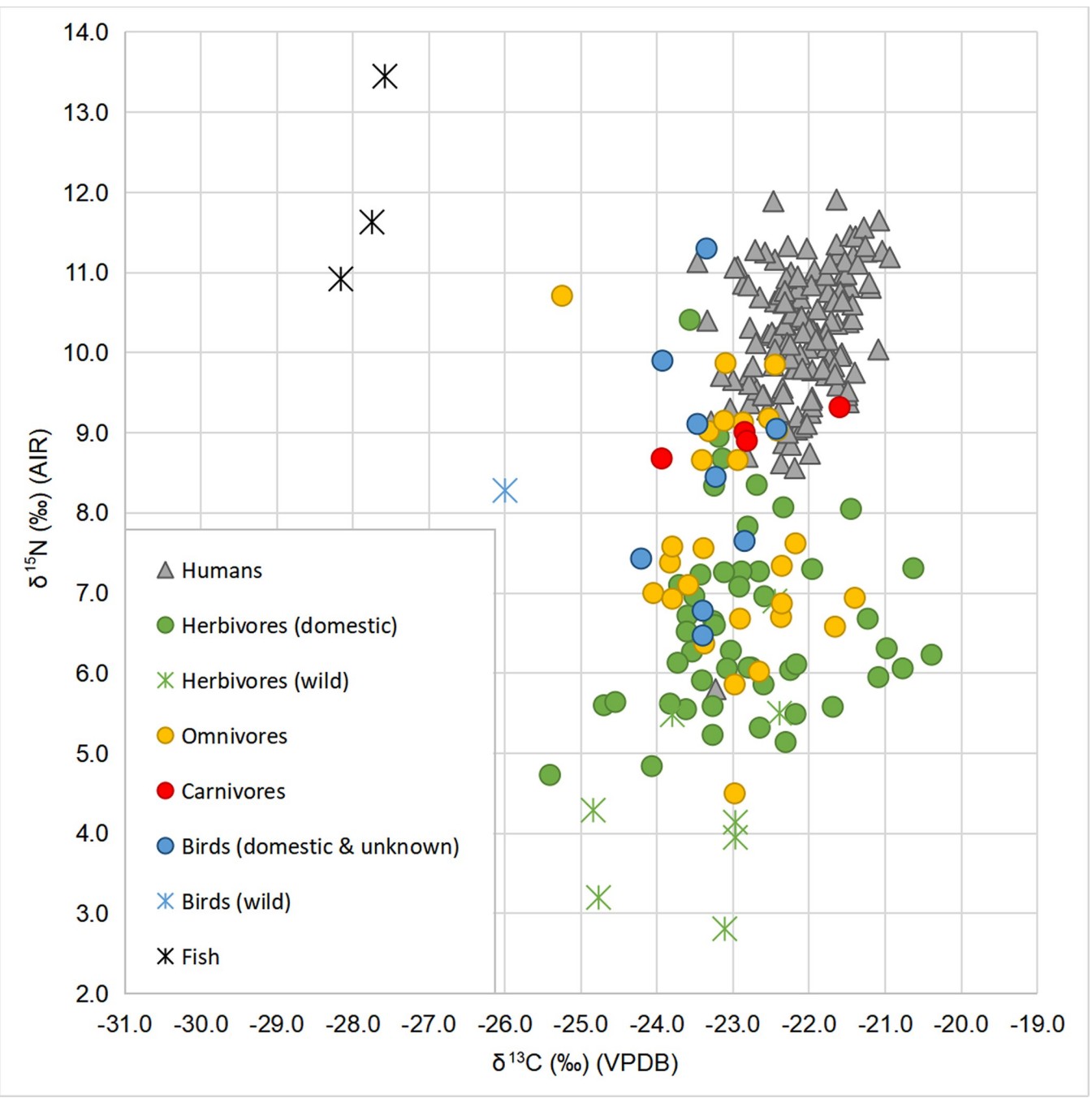

**Fig 2. Plot of δ¹³C and δ¹⁵N values of animals and humans by type of consumer.**

period and 1.4‰ higher than that of the Early Modern period (Fig 3, S5 Table in S1 File). A Kruskal-Wallis test indicates no statistically significant differences in δ¹⁵N values between the time periods (H = 1.433; df = 2; p = 0.565; S6 Table in S1 File). As for the inter-species comparison, while a Kruskal–Wallis test again shows no differences between cattle, sheep/goat and horse/pony in δ¹⁵N values (H = 5.271; df = 2; p = 0.072; S8 Table in S1 File), the horse/pony group is 1‰ lower in δ¹³C than cattle and this difference is significant based on a Kruskal–Wallis test (H = 9.301; df = 2; p = 0.010; S7 Table in S1 File for post-hoc tests).

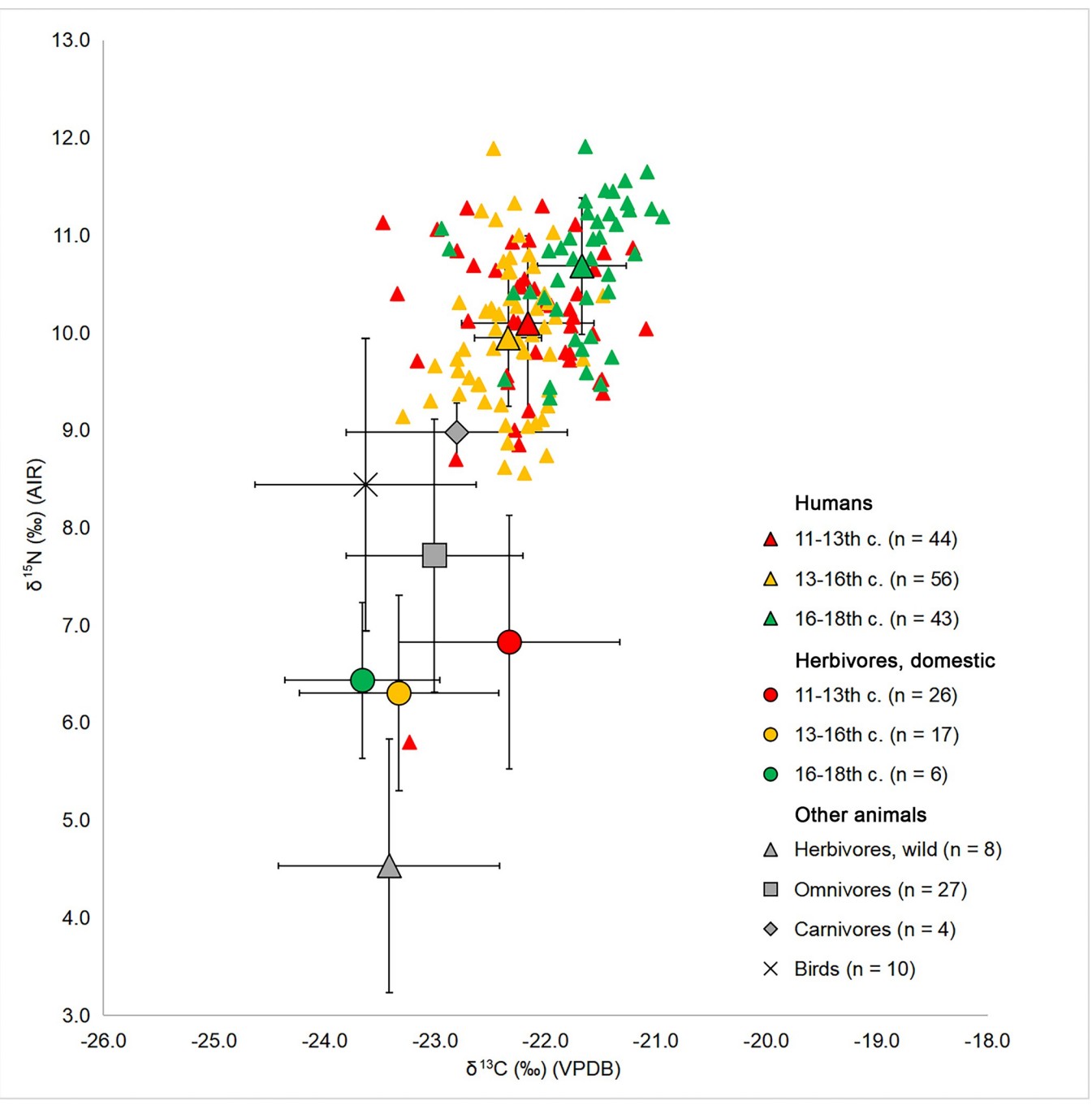

**Fig 3. Plot of δ¹³C and δ¹⁵N values of human and animal samples (mean ± 1 std. dev.) with human and herbivore samples compared by period.** Smaller triangles show the individual values of human samples.

Wild species generally plot lower than the domestic herbivores on the $\delta^{15}N$ axis and cluster more tightly in terms of $\delta^{13}C$. Based on two-sample t-tests, while there are no statistical differences in $\delta^{13}C$ (t = 1.460; df = 55; p = 0.150), the difference in $\delta^{15}N$ between wild and domestic herbivores is significant (t = 4.611; df = 55; p < 0.001) despite the small sample size of wild herbivores. One sheep among the herbivores (Bir-K68.01) exhibits an exceptionally high $\delta^{15}N$ value of 10.4‰.

**Table 4. Summary data for δ$^{13}$C (‰ VPDB) and δ$^{15}$N (‰ AIR) of humans analysed in this study displayed by settlement type, period groups, burial context, sex.**

| | n | δ$^{13}$C Range | δ$^{13}$C Mean | δ$^{13}$C 1 SD | δ$^{13}$C Median | δ$^{15}$N Range | δ$^{15}$N Mean | δ$^{15}$N 1 SD | δ$^{15}$N Median |
|---|---|---|---|---|---|---|---|---|---|
| **Total rural** | 81 | -23.5 to -21.1 | -22.2 | 0.5 | -22.2 | 8.6 to 11.9 | 10.0 | 0.8 | 10.1 |
| *Rural 11-13$^{th}$ centuries* | 23 | -23.5 to -21.5 | -22.2 | 0.6 | -22.2 | 8.7 to 11.3 | 10.1 | 0.7 | 10.1 |
| *Rural 13-16$^{th}$ centuries* | 38 | -23.3 to -21.5 | -22.3 | 0.4 | -22.4 | 8.6 to 11.9 | 9.8 | 0.7 | 9.7 |
| *Rural 16-18$^{th}$ centuries* | 20 | -22.9 to -21.1 | -21.7 | 0.4 | -21.7 | 9.3 to 11.9 | 10.6 | 0.7 | 10.5 |
| **Total urban** | 62 | -23.2 to -20.9 | -22.0 | 0.5 | -22.0 | 5.8 to 11.6 | 10.4 | 0.9 | 10.6 |
| *Urban 11-13$^{th}$ centuries* | 21 | -23.2 to 21.1 | -22.1 | 0.5 | -22.1 | 5.8 to 11.3 | 10.1 | 1.1 | 10.2 |
| *Urban 13-16$^{th}$ centuries* | 18 | -23.0 to -21.9 | -22.3 | 0.3 | -22.3 | 9.1 to 11.3 | 10.3 | 0.6 | 10.3 |
| *Urban 16-18$^{th}$ centuries* | 23 | -22.9 to -20.9 | -21.6 | 0.4 | 21.6 | 9.5 to 11.6 | 10.8 | 0.6 | 10.9 |
| *Elite* | 13 | -22.4 to -20.9 | -21.6 | 0.4 | -21.6 | 9.5 to 11.6 | 10.6 | 0.7 | 10.8 |
| *Military* | 10 | -22.9 to -21.0 | -21.6 | 0.5 | -21.5 | 10.4 to 11.5 | 11.0 | 0.4 | 11.2 |
| **Total 11-13$^{th}$ centuries** | 44 | -23.5 to -21.1 | -22.2 | 0.6 | -22.2 | 5.8 to 11.3 | 10.1 | 0.9 | 10.2 |
| **Total 13-16$^{th}$ centuries** | 56 | -23.3 to -21.5 | -22.3 | 0.3 | -22.3 | 8.6 to 11.9 | 9.9 | 0.7 | 9.9 |
| **Total 16-18$^{th}$ centuries** | 43 | -22.9 to -20.9 | -21.7 | 0.4 | -21.6 | 9.3 to 11.9 | 10.7 | 0.7 | 10.8 |
| **Total male** | 67 | -23.5 to -21.0 | -22.1 | 0.5 | -22.0 | 8.6 to 11.9 | 10.3 | 0.7 | 10.4 |
| **Total female** | 61 | -23.2 to -20.9 | -22.0 | 0.5 | -22.2 | 8.6 to 11.9 | 10.2 | 0.8 | 10.2 |

## 4.2 Human remains

Human bone and dentin collagen (n = 143) has δ$^{13}$C ranging between -23.5‰ to −20.9‰ (mean = −22.1‰ ± 1.0) and δ$^{15}$N ranging between 5.8‰ and 11.9‰ (mean = 10.2‰ ± 1.7). No differences were observed between the bone and dentin collagen samples for either δ$^{15}$N (Mann-Whitney U test, U = 1778; p = 0,102) or δ$^{13}$C (Student's two-sample t-test, t = 0.153; df = 141; p = 0.879). S1 Table in S1 File displays the complete data while Table 4 provides a summary of the average values, their range, and standard deviation. Overall, human values cluster rather tightly above the animals (including carnivores) on the δ$^{15}$N axis, apart from one sample (PUC20-G1) which has an exceptionally low δ$^{15}$N value of 5.8‰ (Fig 2). If this value is excluded, the human range becomes -23.5‰ to −20.9‰ for δ$^{13}$C (mean = −22.1‰ ± 1.0) and δ$^{15}$N distance tightens to a range from 8.6‰ to 11.9‰ (mean = 10.3‰ ± 1.5). A linear regression model of δ$^{13}$C and δ$^{15}$N showed a very weak positive correlation between these parameters for the sample of all humans (R = 0.33, p < 0.001), thus $^{13}$C-enriched fish (either marine or anadromous) is not likely to have been a major source of protein in the diet.

Human (n = 143) mean δ$^{13}$C values for periods of the 11-13$^{th}$, 13-16$^{th}$, and 16-18$^{th}$ centuries are −22.1‰ ± 1.0, −22.3‰ ± 0.6 and −21.7‰ ± 0.8 respectively. Mean δ$^{15}$N values from the earliest to the latest period are 10.2‰ ± 1.4, 9.9‰ ± 1.4, and 10.7‰ ± 1.4. For both δ$^{13}$C and δ$^{15}$N, the values from the latest period of the 16-18$^{th}$ centuries are statistically different from both of the earlier two periods based on Kruskal–Wallis tests (δ$^{13}$C: H = 47.696; df = 2; p < 0.001; δ$^{15}$N: H = 24.276; df = 2; p < 0.001; see S9 and S10 Tables in S1 File for post-hoc tests' results). The comparison of human bone collagen δ$^{15}$N between the urban cemeteries of Polack (n = 62) and the pooled sample of rural sites (n = 81) displays a significantly higher value (10.4‰ ± 1.7 compared to 10.0‰ ± 1.6) for the urban group (Mann-Whitney U test, U = 1651; p < 0.001), but no difference in δ$^{13}$C values (Student's two-sample t-test, t = 1.894; df = 141; p = 0.06). However, the difference in δ$^{15}$N only remains significant in the sample dated to the 13-16$^{th}$ centuries AD (Student's two-sample t-test, t = 2.766; df = 54; p = 0.008; Hedges' g = 0.8) when urban and rural groups are compared separately within each period.

The ranges of δ$^{13}$C and δ$^{15}$N within the nobility sub-sample (n = 13) are from −22.4‰ to −20.9‰ (mean = −21.6‰ ± 0.7) and from 9.5‰ to 11.6‰ (mean = 10.6‰ ± 1.4), respectively (Table 4). The group of individuals identified as potentially having military affiliations (n = 10)

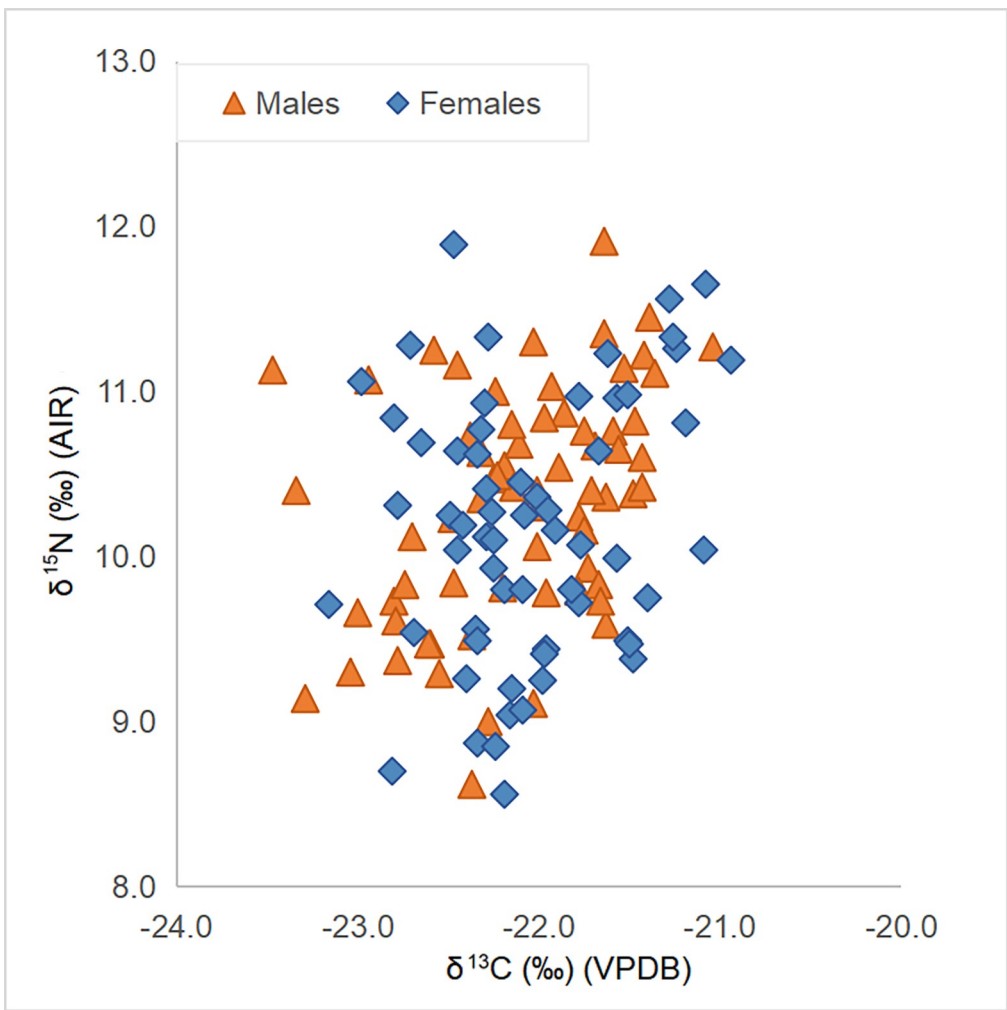

**Fig 4. Plot of male and female δ¹³C and δ¹⁵N values.**

has δ¹³C and δ¹⁵N measurements ranging from -22.9‰ to −21.0‰ (mean = −21.6‰ ± 1.0) and from 10.4‰ to 11.5‰ (mean = 11.0‰ ± 0.8), respectively. The elite sub-sample thus has slightly higher δ¹³C and lower, more variable δ¹⁵N than the military sub-sample. The non-elite (rural) individuals from the 16-18th centuries (n = 20) have δ¹³C and δ¹⁵N measurements ranging from -22.9‰ to −21.1‰ (mean = −21.7‰ ± 0.8) and from 9.3‰ to 11.9‰ (mean = 10.6‰ ± 1.4), respectively. Given that some temporal changes were already detected, status-based groups were compared only to coeval commoners. Elite (n = 13), military (n = 10) and rural (n = 20) human groups from the same time period of the 16-18th centuries showed no statistically significant differences either in δ¹³C (Kruskal–Wallis test, H = 1.840; df = 2; p = 0.398; S11 Table in S1 File) or in δ¹⁵N (one-way ANOVA test, F = 2.061; df = 2; p = 0.141; S12 Table in S1 File). However, a linear regression model applied separately to the elite subsample (n = 13) did show some correlation between carbon and nitrogen values (R = 0.619, p = 0.024; S7 Fig in S1 File).

Males (n = 67) and females (n = 61) present mean δ¹³C values of −22.1‰ ± 1.1 and −22.0 ± 1.0‰ and mean δ¹⁵N values of 10.3‰ ± 1.4 and 10.2‰ ± 1.6, respectively (Table 4 and Fig 4). The t-tests comparing sexes showed no evidence for statistically significant

differences for either $\delta^{15}N$ (t = 1.046; df = 126; p = 0.297) or $\delta^{13}C$ (t = 0.799; df = 126; p = 0.426); this result remains true if individuals with postcranial elements (male n = 53; female n = 40) are tested separately ($\delta^{13}C$: t = 1.212; df = 91; p = 0.229; $\delta^{15}N$: t = 0.439; df = 91; p = 0.662). Likewise, no differences are found between males and females buried in urban or rural cemeteries, or between sexes compared in each period separately.

## 5. Discussion

### 5.1 Diet in Belarus between the 11th and 18th centuries AD

The terrestrial animal data presented here show $\delta^{13}C$ values typical for those feeding in $C_3$ environments. A comparison of domestic omnivore (pig) $\delta^{13}C$ and $\delta^{15}N$ values between sites and chronological periods indicates no statistically significant differences, while the domestic herbivore sub-sample showed a slight, statistically significant increase in $\delta^{13}C$ between the groups of sites dated to the 11-13th centuries AD and the other two periods. Variation in environment or human management practices (e.g. millet foddering) between the geographic locations of sites (Biruli and especially Mienka are located to the south-west of the cluster of Polack, Lučna and Čarscviady, and have higher $\delta^{13}C$ values) or between the periods to which they date are among the possible explanations for this difference. However, more detailed study would be required to determine the reason with higher certainty. As for the 1‰ difference in $\delta^{13}C$ between horse/pony and cattle, similar lower $\delta^{13}C$ values in horses in comparison to cattle and sheep/goats were found in other studies and interpreted as related to differences in pasturing strategies between species or as potentially caused by metabolic processes [94–96]. This difference may contribute to the higher $\delta^{13}C$ values observed among domestic fauna in the 11-13th centuries, as the proportion of fauna represented by cattle is the biggest for this period (n = 18 compared to 8 in 13-16th centuries and 2 in the 16-18th centuries), while both horse and cattle samples are rather evenly distributed among sites. Rural herbivores have a higher $\delta^{15}N$, possibly due to obtaining more of their fodder from manured fields. Though we have previously discussed that city inhabitants increasingly obtained their food from rural neighbourhoods, agricultural activity was common in cities in the 10-16th centuries and city inhabitants also kept their own gardens and domestic animals [7,9,29,30,33,37,41], which may explain why there are differences between the animals from rural and urban contexts. Interestingly, the sheep Bir-K68.01, which exhibits a $\delta^{15}N$ value of 10.4‰, was found in a kurgan. It is thus the only animal in the dataset deposited in a human burial rather than a settlement context and is represented not by a single fragment but rather part of a whole skeleton. Such enrichment in $^{15}N$ may be due to extreme protein stress like starvation, or being fed a specific diet, or perhaps due to this individual coming from some distance away where local baseline $\delta^{15}N$ values were higher due to manuring [15]. Given the age-at-death of the animal is at least 1.5 years, the weaning effect is unlikely. Overall, the faunal data shows little change in the baseline $\delta^{13}C$ and $\delta^{15}N$ values across space and time, with the exception of an increase in the 11th-13th centuries AD when compared to the other temporal groupings, providing a useful baseline for interpreting human diets. The aforementioned decrease in the domestic herbivore values following the 11-13th centuries' period will be accounted for in the analysis of human results.

Given that no differences were observed between the isotope values in the bone and dentin collagen samples, and the results of male to female comparison stay the same for both the entire sample and for individuals with postcranial elements when they are tested separately, we assume that neither the heterogeneity of samples nor the variability of methods applied for sexing (based on cranium or postcranial elements) have substantially affected the interpretation of the results.

Most of the human samples fall within the range of -23.5‰ to −20.9‰, suggesting a clear reliance on $C_3$ terrestrial protein, which is consistent with the expected diets of Medieval European populations dominated by $C_3$ grains along with vegetables [97], and with the agricultural system prevalent on Belarusian lands in particular. The data in this study do not indicate the significant presence of millets in the diet in general, similar to what has been found by existing archaeological research [29,30]. Human $\delta^{15}N$ values consistent with high trophic position, especially among the individuals from rural graveyards, are somewhat contradictory to the proposition that little meat and milk products were consumed by commoners [26,32]. Given that the classic picture of meat appearing on the peasants' table on festive occasions was drawn primarily from ethnographic sources [26,31,32], which only relate to the 19-20th centuries, our isotopic data might suggest that animal protein before that time was actually more common in the diet of commoners. This is more in line with contemporary sources from neighbouring Medieval Poland, where meat was supposedly considered a basic food to be consumed on a daily basis even among the general populace [98]. A notable exception among human individuals is the outlier PUC20-G1 with an exceptionally low $\delta^{15}N$ value of 5.8‰. This individual from Polack in the 11-13th centuries AD plots at the lower end of the herbivorous nitrogen values, which implies a vegetarian diet or substantial reliance on legumes [99]. Domestic species were likely the major source of animal protein for the majority of the population, as the offset between human $\delta^{15}N$ values and those of the wild herbivores is more than 5‰ and thus outside the expected trophic level increase. The fact that wild herbivores are on average 2.1‰ lower than domestic herbivores in terms of their $\delta^{15}N$ values is likely connected to domestic animals grazing on manured fields. Our data thus supports that manuring was likely utilised in the crop-rotation agricultural system at the time, also in the north of Belarus. Differences between species may also have some effect on the lower nitrogen values (e.g. the proportionally higher number of small herbivores from Lagomorpha and Rodentia orders versus the Artiodactyla in the wild animal sample), though there is no strong pattern between wild species and isotopic values at the intra- or inter-species level. Although wild fauna do have lower $\delta^{13}C$ values than domesticated fauna, this difference is not statistically significant. Furthermore, our small sample of wild fauna includes both species from woodland and near woodland environments (e.g. red deer, bison, elk) and grassland or wetland (e.g. hare, beaver, wild bird). As a result, there does not seem to be an obvious impact of the canopy effect in our study, though larger analyses in the future could explore this further.

Another theme that was widely debated in terms of Medieval temporal trends was the adoption and growing role of Christianity, which in dietary terms would be expected to be reflected in a popularity of fish consumption [2,4,5,42]. The data in this study do not support any significant reliance on aquatic resources, despite the assertion that fasts were kept very strictly in these lands [32], despite water resources being abundant in the north of Belarus [7,40], and despite fishing equipment appearing among archaeological finds here [39]. A similar low role of fish despite expectations connected with Christian fasts and the local availability of fish was observed in other non-elite sites, such as Alytus, Gruczno or Kałdus, as well as the coastal rural sites of Kretinga and Palanga in Lithuania [42]. The general lack of evidence for fish consumption may mean that the inhabitants of Belarusian lands were not so strict in observing Christian fasts or were participating in them in a different way than through focusing on fish consumption (e.g. nuts or berries, waterfowl and aquatic animals like beaver, which were treated as fish-suitable for fast). However, freshwater fish are difficult to detect isotopically due to their wide ranges of $\delta^{13}C$ and $\delta^{15}N$ values, and $\delta^{13}C$ values of the human samples from Belarus do overlap both the anadromous fish ranges of -24‰ to -12‰ suggested in the literature [2] and $\delta^{13}C$ values of some of the Polish and Lithuanian fish samples, such as vimba, catfish,

carp-bream and pike-perch—species caught in Belarus as well. Therefore, consumption of freshwater and anadromous fish cannot be completely excluded based on these data, either.

No evidence for dietary variation between males and females was observed, either in the whole sample or within its subgroups based on urban or rural context or time period. Furthermore, the sub-sample of contemporary nobility from Polack showed no significant differences to rural sample in terms of $\delta^{15}N$ values, contrary to the proposition of a drastic increase in the consumption of animal protein by the nobility after the 16th century under the Polish influence [48]. The only clear difference that the elite sub-sample shows relative to the rest of the individuals is a moderate correlation between the carbon and nitrogen values, perhaps suggesting consumption of anadromous or marine fish did play some role in elite isotopic variation. However, overall the lack of a significant increase in $\delta^{15}N$ value along social status lines suggests limited differences in animal protein access. A possible explanation could be that the nobility were consuming greater amounts of wild animal protein (which seems to overall have lower $\delta^{15}N$ values in our study), which could offset increased animal protein use among elites and further highlight the significance of hunting as a sign of status in the region. Alternatively, limited variation in animal protein consumption between elites and non-elites could emphasise the significance of the Christian fasting tradition, particularly for the Catholic confession to which this group of deceased is believed to belong, and use of fish and plants in the diet. The lack of expected differences between this group and the rural subsample could be explained by a lack of corresponding lived differences in terms of social status in the analysed individuals. Nevertheless, if we assume that the archaeological interpretation of social distinction is correct, then our results either show i) that commoners had similar access to status foods such as animal protein as the 'elite' or ii) that the elite represented come from a poorer portion of the GDL's aristocracy. Interestingly, the military sub-sample does seem to have higher $\delta^{15}N$ values than members of the elite, without any corresponding $\delta^{13}C$ trend, suggesting greater access to domestic animal protein or freshwater fish.

In terms of the comparison between city and village populations, we can see that urban or rural settings did not play too much of a major role in determining diet, at least at the resolution of isotopic analysis. Polack inhabitants have $\delta^{15}N$ values on average 0.4‰ higher than villagers. This small difference is significant only in the 13-16th centuries; however, the effect size of this difference for the 13-16th centuries is large (Hedges' g = 0.8) and it is notable that at least the urban herbivores at this time have lower $\delta^{15}N$ values. A possible explanation is a slightly higher proportion of animal or freshwater fish protein in the urban diet, which could have been more available in such an important economic centre as Polack. The reason why the difference is visible only between the 13th and 16th centuries may be due to the fact that prior to this point the differentiation between the town and countryside was yet emerging, with the majority of city dwellers engaged in agriculture and husbandry [9]. The lack of significant difference between rural and urban samples in the 16th - 18th centuries may reflect the economic decline of Polack due to constant military conflicts and raids on northern Belarus by Russian troops that became regular since the 1510s, and in particular the Livonian War of the second half of the 16th century—all of which resulted in the influx of rural population [9], potentially contributing to the observed similarity between the diets of the inhabitants of Polack and the surrounding countryside. Despite the initial proposition that urban diets could be more heterogeneous due to obtaining food from markets where not only strictly local but more distant and even foreign foods could be available, urban dwellers seem to, overall, cluster rather tightly, in a similar manner to the rural communities.

One of the major aims of this study was to investigate dietary change through time. In this regard, the only significant change was a slight increase in both carbon and nitrogen values after the 16th century. This suggests that, overall, the emergence of new states had little impact

on the ordinary people, especially in rural areas. However, as the change seen following the 16th century cannot be explained by other parameters such as the environmental baseline (it showed a different pattern) or the variable nature of the samples dated to various periods (a similar pattern is preserved if rural and urban or male and female groups of individuals are compared separately by period), it is important to discuss what factors could have caused this shift. If the $\delta^{15}N$ values are simplistically interpreted as greater access to a higher trophic or aquatic protein source, then the slight increase of them in the time of the Polish-Lithuanian Commonwealth would be in line with the picture of the "Golden Age", not only for the highest strata of the society but also more broadly across the population. The simultaneous increase in $\delta^{13}C$ values in the period following the formation of the Polish-Lithuanian Commonwealth may also reflect the adoption of $^{13}C$-enriched fish into the diet by part of the population, which would be consistent both with the reported adoption of some marine imports into the menu of the nobility and with the striking reliance on such fish reported by Reitsema et al. [2] among Polish nobility of the same time period. Finally, the increase in human $\delta^{13}C$ values could indicate higher millet consumption in this period. This would perhaps fit with increased reliance on millet during the adverse weather events of 1600–1603 and the influence of the Little Ice Age, and the aforementioned constant military conflicts starting from the early 16th century, which destroyed harvests among other things and undermined the economic contributions of the territory [34,100]. That said, the corresponding change in $\delta^{15}N$ would perhaps imply that aquatic resources are the more likely contributors. Overall, human diets remain remarkably consistent between social groups and between time periods from an isotopic perspective.

## 5.2 Comparison with the surrounding region

We also compared our data with other geographically and chronologically proximate datasets available in the literature, namely from Poland [2,3] and Lithuania [5,42]—two countries with which Belarus formed a common state at different points in history. Unfortunately, the raw data from Yaroslavl [51] was not provided in the publication, thus statistical comparison to the Russian territory is not possible, limiting us to general observations in this instance. The isotopic values for animals from all of the studies are shown in Fig 5. Only domestic herbivores had enough representation in studies from all three countries to allow for meaningful comparison through statistical tests. Based on Kruskal–Wallis tests, no difference was observed in $\delta^{15}N$ (H = 1.261; df = 2; p = 0.532; S14 Table in S1 File), while stable carbon isotope values differed significantly between Belarus and the other two countries (H = 23.004; df = 2; p < 0.001; refer to S13 Table in S1 File for the results of the post-hoc tests). The means of domestic herbivores from Belarus (n = 49), Lithuania (n = 17), and Poland (n = 8) were -22.8‰ ± 2.1, -22.1‰ ± 0.8, and -21.3‰ ± 0.9, respectively. As can be seen from Fig 5, omnivores, carnivores, and even birds and wild herbivores repeat the pattern where $\delta^{13}C$ values increase from Belarus, to Lithuania, to Poland, and it is statistically confirmed if the pooled samples of animals (excluding fish and marine mammals) are compared across these regions.

Given that even wild herbivores exhibit this pattern, it is unlikely to be explained by human intervention. More plausible here is a difference in the environmental baseline affecting all animals, stemming from the plants consumed by herbivores and maintained along the whole trophic chain. Various characteristics of the local environment, such as water stress, precipitation, humidity, temperature, number of hours of sunshine, and salinity of soil influence $^{13}C$ abundance in $C_3$ plants [101–104], causing 3–6‰ variation in their $\delta^{13}C$ values [101,104]. In particular, Somerville et al. [104] found a significant correlation between temperature and $\delta^{13}C$ values in the bone collagen of rabbits and hares, which in $C_3$-plant dominated

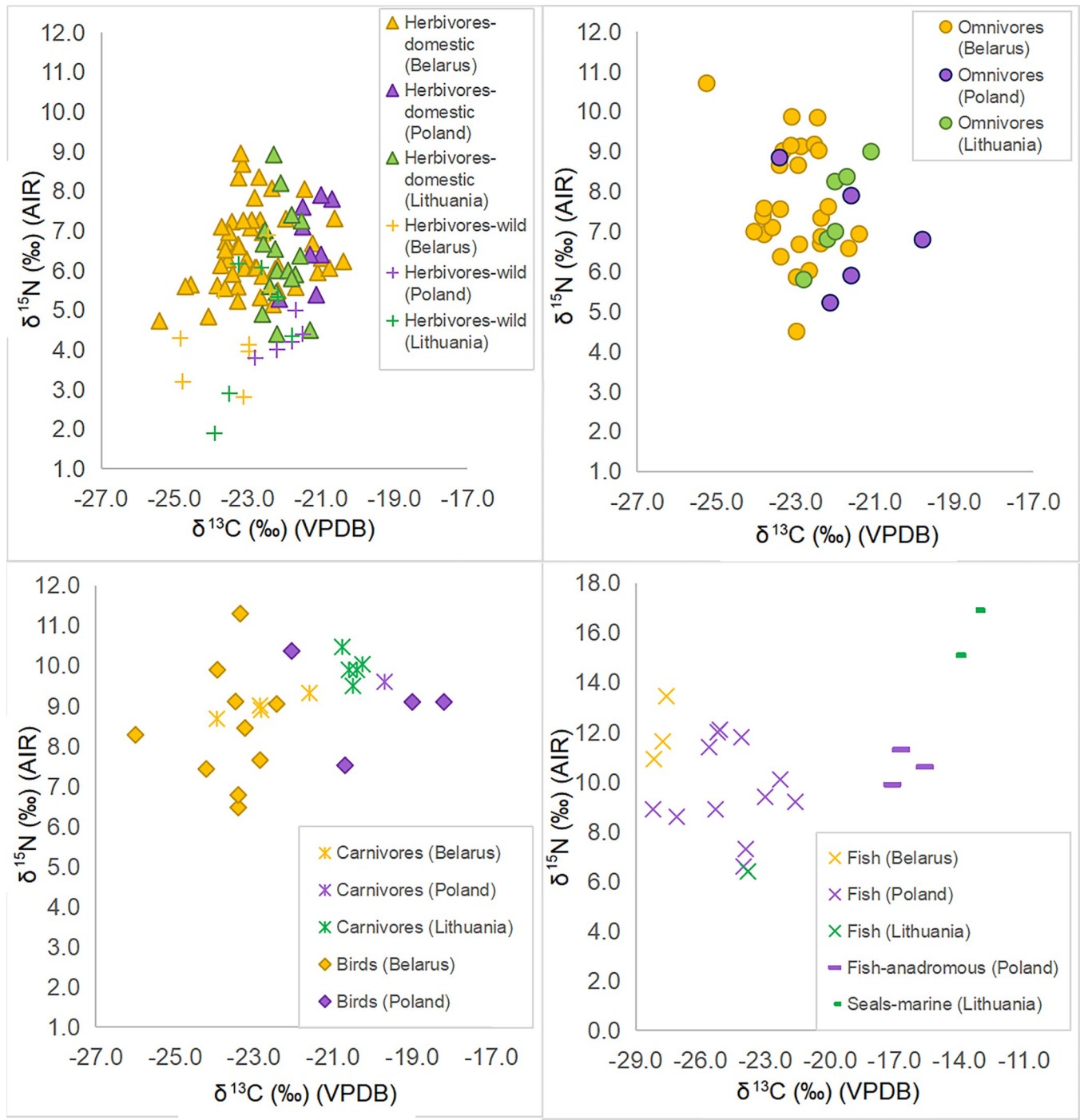

**Fig 5. Plots of δ¹³C and δ¹⁵N values of herbivores, omnivores, carnivores with birds, and fish from the three countries (Belarus, Lithuania, Poland).**

environments may be due to $C_3$ plants exhibiting higher $\delta^{13}C$ values when water stressed relative to plants from cooler and wetter regions. This pattern seems to fit with the differences between countries compared in the current study, with Belarus being cooler and having more wet days than Lithuania, and Lithuania than Poland based on modern climatic data [105].

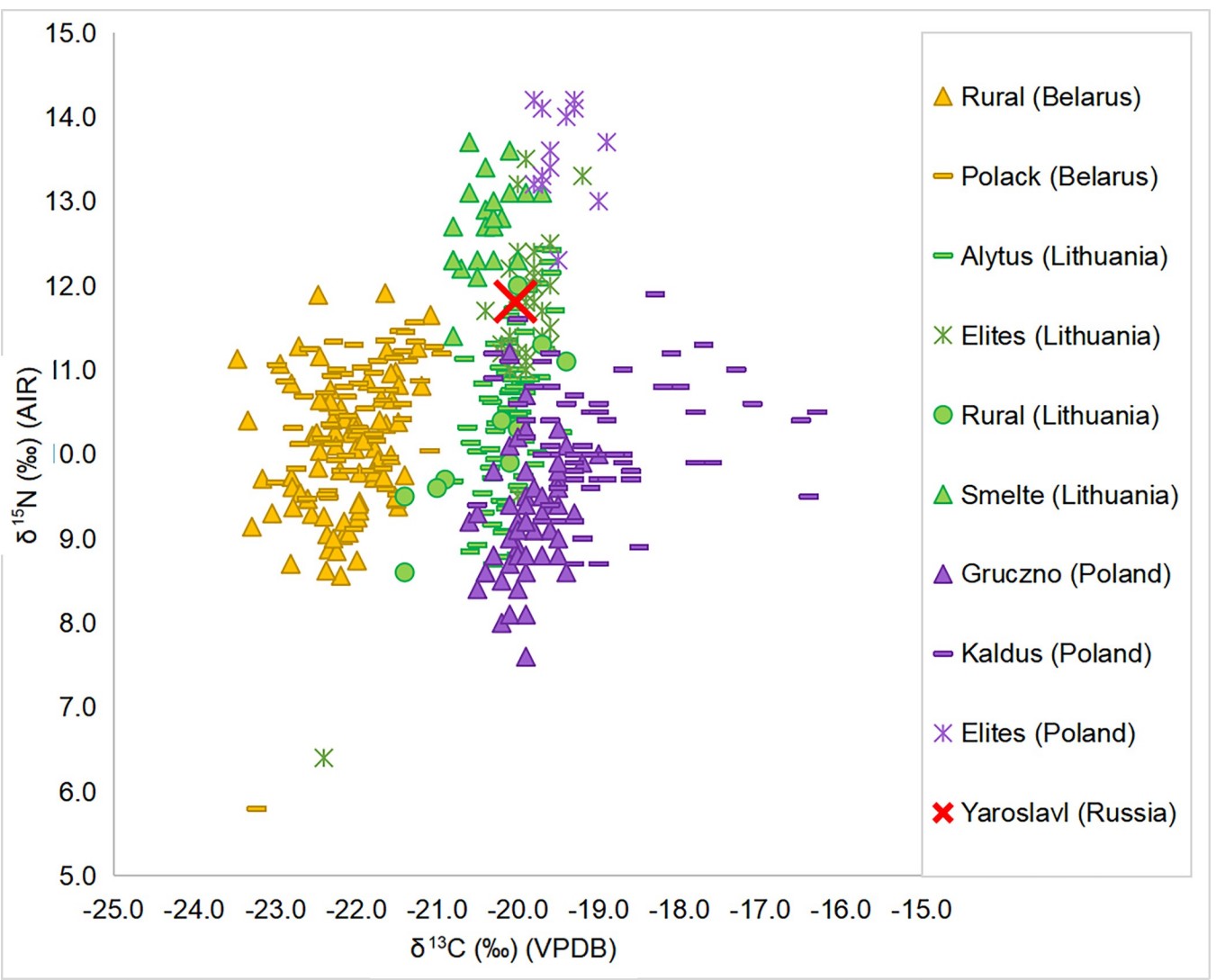

**Fig 6. Plot of δ¹³C and δ¹⁵N values of human samples from sites in Belarus, Lithuania and Poland.**

In terms of human data, all Belarusian sites (both rural and Polack) cluster quite distinctly from the Polish and Lithuanian data (Fig 6). In terms of $\delta^{13}C$ values, Kruskal-Wallis multiple comparisons show that Belarusian sites are significantly different to the others. Kałdus, a Medieval economic hub in Poland, is likewise separated from all but the Polish elites while other sites largely fall between them within a range of -21‰ and -19‰ (see S15 Table in S1 File for the post-hoc tests' results). The $\delta^{13}C$ distinction noted at Kałdus is likely due to substantial evidence for millet consumption reported based on stable isotope analyses [3]. Fish consumption was considered another contributing factor to higher $\delta^{13}C$ values, with fish providing a ready solution to the problem of feeding agglomerated populations detached from farming [3]. Human $\delta^{15}N$ across the different regions tends to be about one trophic level higher than associated animals, indicating significant consumption of terrestrial animal protein. Kruskal-Wallis tests indicate that Belarusian rural sites and Polack have similar $\delta^{15}N$ to the Polish site of Kałdus (p = 1.000), Lithuanian rural sites (p = 1.000) and Alytus (p = 0.313 for Belarusian rural sites and p = 1.000 for Polack sites; see S16 Table in S1 File for the post-

hoc tests' results). Kaldus was an economic centre of the 10-13[th] centuries AD in Poland where inhabitants both produced and bought part of their food and where the animal protein in the diet came from domestic terrestrial animals complemented by fish [3]. Alytus was a small inland Medieval town in Lithuania, where most citizens engaged in agrarian occupations, such as agriculture, hunting, fishing, and animal husbandry, having a homogeneous diet based on $C_3$ terrestrial resources such as wheat, barley, cow, and pig [5]. Lithuanian rural sites refer to Medieval–Early Modern inland Kretinga and coastal Palanga, which both showed a high reliance on terrestrial resources and a diet based on cereals and legume crops and plant gathering (mostly mushrooms and nuts), supplemented by pork fat [42].

Lithuanian and in particular Polish elites, as well as the coastal site of Smeltė, stand out most distinctly from the other sites with higher $\delta^{15}N$ values. For the elites, this is explained by increased access to animal protein and freshwater and migratory fish consumption, which could have been connected to the elite consumer status of many fish (e.g. sturgeon), as well as the fact that many of these elites were related to the clergy [2]. This pattern is only partially repeated by the small group from Polack interpreted as burials of city nobility, who show some evidence of $^{13}C$-enriched fish in the diet but no difference in the mean $\delta^{15}N$ values compared to the non-elite individuals. Based on Kruskal–Wallis tests, the sub-sample of nobility from Polack (mean $\delta^{13}C$ = -21.6‰; mean $\delta^{15}N$ = 10.6‰) is significantly lower in $\delta^{13}C$ and $\delta^{15}N$ than the Lithuanian (mean $\delta^{13}C$ = -20.0‰; mean $\delta^{15}N$ = 11.6‰) and especially Polish elites (mean $\delta^{13}C$ = -19.5‰; mean $\delta^{15}N$ = 13.6‰) from church burials ($\delta^{13}C$: H = 50.920; df = 3; p < 0.001; $\delta^{15}N$: H = 42.290; df = 3; p < 0.001; refer to S17 and S18 Tables in S1 File for the post-hoc tests' results). In this regard it should be noted that nobility in the Grand Duchy of Lithuania ranged from rather poor aristocracy to the most powerful and richest stratum of magnates who must have differed drastically in their lifestyles and diets. Unlike the Polish and Lithuanian elite samples, the sample considered to be nobility from Polack comes from a burial ground in the city centre. Relatively higher reliance on hunting and consumption of wild animal' meat on Belarusian lands could at least partially account for the lower nitrogen values of Polack nobility. Interestingly, according to Gurevich [47], hunting played an exceptional role in the cities of Black Rus, and in Navahrudak specifically—36–58% of animal bones found there belong to wild species. However, such a significant role of hunting is not necessarily true for times after the Ancient Rus and for other cities of the Belarusian lands.

The non-elite samples from all regions do not show any clear evidence for fish consumption, apart from the modern coastal site of Smeltė in Lithuania, where diets seem to have included both terrestrial protein and considerable amounts of protein from freshwater fish. The 13[th]-century Russian city of Yaroslavl with its mean values of -20.0‰ for $\delta^{13}C$ and 11.8‰ for $\delta^{15}N$ (Fig 6) was interpreted as demonstrating mixed diets based on plants from a temperate climate and meat from local terrestrial herbivores, where rare individuals with $\delta^{15}N$ values exceeding 13‰ probably regularly consumed freshwater fish [51]. This sample would plot much higher in both isotopic values than the Belarusian sites, and closest to the cluster of Lithuanian elites. It is particularly noteworthy given that the sample from Yaroslavl represents the full population of a 13[th]-century city, not an elite subgroup from it. Overall, based on the examined samples, diets seem to have been dominated by $C_3$ plants and animal resources across this part of eastern Europe between the 11[th] and 18[th] centuries AD. Fish seems to have played a more marginal role than expected based on Christian fasting tradition, and was more easily accessed in urban or elite contexts unless rural sites were on the coast. In terms of elite groups, those from Belarus have lower $\delta^{15}N$ than those analysed in Poland or Lithuania, perhaps indicating more similar diets to commoners than expected or the hunting and consumption of wild animals as a source of status. Future work, including single compound amino acid analysis, may help to definitively resolve this. The observed pattern does, however, stand in

stark contrast to datasets from Medieval western Europe where clear isotopic distinctions between different elite and urban groups, and commoners, have been observed [4].

## 6. Conclusions

We present a new stable isotope dataset, assembled to explore how diets in the Polack region of Belarus changed throughout the 11-18th centuries in urban and rural settings, between male and female individuals, and across different burial settings. Overall, as also seen from a comparison of data from the wider eastern European region, the diet in Medieval Belarus was based on $C_3$ terrestrial resources, among which cereals like rye and wheat, as well as vegetables played the major role. On average, high access to animal protein for commoners is also observed. In general, the type of site (urban or rural) does not seem to have been a primary factor determining diet, with differences mainly centred around access to fish in inland settings. The differences in diets between elite and non-elite individuals are similarly limited and probably due to religion or status-induced fish consumption across society, a higher role of hunting in the Belarusian context, or maybe reflecting the broader variety of lived experiences that the simplistic distinctions like elite versus commoner are not reflecting. Although more work is needed to further investigate the patterns observed in this study, our stable isotope analysis demonstrates how historical record can be tested in Medieval contexts, with eastern Europe providing an interesting and yet under-investigated comparison to other areas of Medieval Europe. Future research should focus on a more comprehensive reconstruction of dietary behaviours and variation by including individuals from broader geographic regions and time periods, as well as representing different strata of the society, such as magnates, clergy and individuals belonging to different confessions. Nevertheless, the data provides an important counterpoint to existing research from western Europe that appears to demonstrate more stark differences between rural and urban settings and between elite individuals and commoners [4].

## Supporting information

**S1 File. Supplementary S1-S3 Notes, S1-S7 Figs, S1-S18 Tables and supplementary references.**
(DOCX)

## Acknowledgments

The authors express gratitude to Dr. Urszula Iwaszczuk (Institute of Mediterranean and Oriental Cultures, Polish Academy of Sciences) for her help with the identification of faunal species; to professor Arkadiusz Sołtysiak (Warsaw University) for supervision and advice; to D.S. Gritskevich and V.A. Yemieljanchyk from Polack State University (PSU) for determining the biological profiles of individuals from their university's osteological collection, as well as continuous aid in sample acquisition; to M.M. Pamazanau and O.V. Marfina from the Department of Anthropology, Institute of History of the National Academy of Sciences (NAS) of Belarus for help with organisational issues and for the access to materials; to A.V. Voytekhovich (NAS) and M.V. Klimau (NAS) for the provisioning of animal materials and the related information; to Nadzeya Haponava and Mikhail Shatsila for work on the figures; and to V.U. Charauko (PSU) for materials.

## Author Contributions

**Conceptualization:** Vera Haponava, Patrick Roberts.

**Data curation:** Vera Haponava.

**Formal analysis:** Vera Haponava.

**Funding acquisition:** Patrick Roberts.

**Investigation:** Vera Haponava, Max Both.

**Methodology:** Vera Haponava, Mary Lucas, Patrick Roberts.

**Project administration:** Vera Haponava.

**Resources:** Aliaksei Kots, Mary Lucas.

**Supervision:** Patrick Roberts.

**Visualization:** Vera Haponava.

**Writing – original draft:** Vera Haponava.

**Writing – review & editing:** Vera Haponava, Aliaksei Kots, Patrick Roberts.

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
