## [Decision Letter · Decision Letter 0]

19 May 2022

PONE-D-22-08905Medieval and Early Modern diets in the Polack region of Belarus: A stable isotope perspectivePLOS ONE

Dear Dr. Haponava,

Thank you for submitting your manuscript to PLOS ONE. After careful consideration, we feel that it has merit but does not fully meet PLOS ONE’s publication criteria as it currently stands. Therefore, we invite you to submit a revised version of the manuscript that addresses the points raised during the review process.

The manuscript ‘Medieval and Early Modern diets in the Polack region of Belarus: A stable isotope perspective’ reports an interesting dataset on the medieval to Early modern population of the Polack region in Belarus.

The manuscript opens a new window on a portion in Europe where isotopic data on human remains are rare and welcome. However, the manuscript has problems that the Authors have to address before this contribution could be suitable for publication in Plos One.

Both reviewers did a detailed review of the paper, and the authors must address carefully their criticisms.

I have little to add the reviewers’ points:

The archaeological record of the excavated sites is lengthy reported in the SI, but I urge the authors to provide some figures of the excavated graves and/or excavation plans. Unfortunately, almost all the archaeological reports are in Belarus language and difficult for the common reader to understand the differences between the funerary customs (e.g. kurgan vs other grave types) and the archaeological settings. Moreover more details should be provided about how the site chronology were determined.Bioarchaeology of the human remains, sex and age at death methods should be described and justified better. It is hard to believe to sex estimates made on the cranium only of age at death based on cranial sutures analysis. Moreover, the authors write  “In the cases where sex and age estimation based on the study of the cranium contradicted those stated in the available publication or archaeological report, the latter assessments were used based on the assumption that it could have been estimated with more certainty at the time of analysis when postcranial elements were also available”. So, why did the authors a new anthropological analysis of the human remains?

We look forward to receiving your revised manuscript.

Kind regards,

Luca Bondioli, M.D.

Academic Editor

PLOS ONE

Journal Requirements:

2.We note that the grant information you provided in the ‘Funding Information’ and ‘Financial Disclosure’ sections do not match. 

Reviewers' comments:

Reviewer's Responses to Questions

**Comments to the Author**

1. Is the manuscript technically sound, and do the data support the conclusions?

Reviewer #1: Yes

Reviewer #2: Yes

2. Has the statistical analysis been performed appropriately and rigorously? 

Reviewer #1: Yes

Reviewer #2: No

3. Have the authors made all data underlying the findings in their manuscript fully available?

Reviewer #1: Yes

Reviewer #2: Yes

4. Is the manuscript presented in an intelligible fashion and written in standard English?

Reviewer #1: Yes

Reviewer #2: No

5. Review Comments to the Author

Reviewer #1: The paper “Medieval and Early Modern diets in the Polack region of Belarus: A stable isotope perspective” explores patterns of paleodiet of inhabitants of the Polack region – Belarus – across time (11th-18th centuries), between sex, settlement type (rural vs urban) and burial context (elite vs military) through carbon (δ13C) and nitrogen (δ15N) isotope analysis of human and faunal bone and dentine collagen. With its significant set of data of human (n=143) and faunal (n=105) remains from 15 rural graveyards and the town of Polack, the paper constitutes a valuable contribution to reconstructing dietary habits across significant historical periods in Belarus - an under-investigated region compared to other parts of Europe for the reconstruction of past human diets. I find particularly noteworthy the extensive archaeological and historical background given in the Supplementary Information and the integration of various archaeological, literary, and historical sources with δ13C and δ15N outcomes in the discussion. The paper also compares the data with δ13C and δ15N values from neighbouring territories, showing the main differences. The data are exhaustively reported in the Supplementary Tables (Supplementary Tables 1 and 2) and summarised for chronological periods, type of site and type of burials (Tables 3 and 4). Quality control is in place and reported in the Supplementary Tables. The statistical analysis performed is adequate for the study. The large chronological span and diverse type of site (rural vs city) provide the dateset for answering the research questions the authors outline.

However, the manuscript needs some revisions. As a general comment, there are some typos and problems with consistency in how the results are reported. The authors might want to consider making some changes to the Figures. All historical and archaeological bibliography is in Belarusian or Polish language, and it would be useful to have more general references in English. The various sites’ funerary contexts should be better described. The authors might want to clarify how reliable the given chronology is and explain how this latter was defined (e.g., based on archaeology/typology). A major concern is related to the age-at-death determination of adult individuals, which shows very narrow ranges compared to the methodology used. Additionally, in the methods, the authors rightly highlight the difficulties of comparing bones with tooth roots as they reflect different years of life, but then this aspect is never taken into account in the discussion of results. Finally, some parts of the text in the discussion are slightly hard to follow and forced when integrating δ13C and δ15N analysis and historical data.

It follows some minor and major points that I suggest the authors amend:

MAIN TEXT:

MINOR:

- Indentation does not seem consistent across the main text and Supplementary Information [e.g., Lines 103, 251, 299, 314]. The authors might want to consider this aspect throughout the text.

- The spacing when reporting results and samples’ numbers does not appear to be consistent across the text. For example: (n = 49) [Line 401], (n=10) [Line 421], (n=51) [Line 439], (n = 142) [Line 463]; (t=1.894; df=141; p=0.06) [Line 472], (F = 6.381; df = 2; p =0.003) [Lines 438-439], (2.255; df = 2; p = 0.324) [Line 444]. The authors might want to consider this aspect throughout the text.

- E.g. is reported sometimes with a comma, other times without a comma. For example: (e.g., maize and millet) [Line 188] vs (e.g. meat, aquatic) [Lines 195-196].

- Lines 89-91 = The authors might want to give a range for the diverse chronology – “Ancient Rus principalities (9th century AD), the Grand Duchy of Lithuania (13th century ASD) and the Polish-Lithuanian Commonwealth (16th century AD)”.

- Line 90 = Typo – “Lithuania (13th century ASD)”.

- Line 93 = Commoners vs elite – The definition appears a bit simplistic since nothing prohibits that there were individuals with greater status or wealthier among the commoners and vice versa.

- Lines 112-116 = Add reference or refer to SI – “Over the course of the period between the 11th and 18th centuries, these formations included Ancient Rus, the Grand Duchy of Lithuania (GDL), and the Polish-Lithuanian Commonwealth (PLC). The latter state ceased to exist at end of the 18th century and the Polack region, as well as other Belarusian lands, was gradually annexed by the Russian Empire”.

- Lines 114-115 = “at end of the 18th century” – at the end.

- Lines 116-118 = Add reference – “As social identity is believed to be deeply embedded in diet and consumption behaviours, these events might have been paralleled by dietary changes”.

- Line 181 = “Stable isotope analysis and palaeodiet in Medieval eastern Europe” but Poland dataset includes “2nd century AD to 10-18th centuries”.

- Line 232 = “Kurgan” – the authors might want to define the term.

- Line 316 = The authors might want to add what types of other bone tissues were sampled as reported in Supplementary Table 1.

- Line 397 = “Abandoned” – The authors might want to find another word.

- Line 435 = “Insignificant” – The authors might want to find another word.

- Line 448 = “Human bone collagen” – Human bone and dentine collagen.

- Line 463 = Typo – “Human (n = 142) mean δ13C values” – (n = 143).

- Line 531 = Add reference.

- Line 550 = Add reference.

- Lines 592-593 = “Polack inhabitants have δ15N values on average 0.5‰ higher than villagers”– should be 0.4‰ (rural δ15N Mean: 10.0‰; urban δ15N Mean: 10.4‰).

- Lines 661-662 = “As can be seen from Figure 3” – maybe Figure 5.

- Line 759 = “As is also seen” – As it is also seen.

MAJOR:

- Lines 339-340 = In Supplementary Table I, age-at-death is reported with very narrow ranges (e.g., 55-60 or 45-50) mainly if this was estimated based on the cranial suture closure observation only – “The age-at-death was assessed observing cranial suture closure following Piontek [71]”.

- Lines 344-345 = When estimating age-at-death based on dental wear patterns, the authors might want to use (Lovejoy 1985) instead of (Brothwell 1981).

- Lines 531-534 = The authors might want to reconsider this sentence. I am not entirely sure we can relate 19th-20th century ethnographic sources with what happened in the 11th-13th and 13th-16th century.

- Lines 502-507 = The sentence is slightly unclear and imprecise. Additionally, the authors might want to explain better what type of omnivores were considered.

- Lines 590-614 = The interpretation is slightly unclear – maybe the authors want to consider reframing this part and contextualising it more clearly. I also think that the sentence “Considering that this is a more expensive source of energy, it might mirror the important economic role of Polack in reflecting the generally higher living standards and food security of its inhabitants” might be an overstatement since 0.4‰ (general urban vs rural) or 0.5‰ (13th-16th century) is a too little shift.

- Lines 623-647 = When discussing “what factors could have caused this shift” in δ13C and δ15N values after the 16th century, there is a bit of overreaching when explaining the possible increase in millet consumption. The authors might want to reconsider this aspect.

FIGURES:

- Fig. 1 Legend – The authors might want to consider adding in the legend the dates in the format reported in the text (e.g., 11-13th AD). The authors might want to consider spacing the legend animal-period from human-period a bit more.

- Fig. 3 Caption – The authors might want to consider inserting δ13C and δ15N (e.g., Plot of δ13C and δ15N values of human and animal samples compared by period).

- Fig. 3 Legend – The figure appears slightly hard to read – the authors might want to consider giving the same colour for each period and the same shape for each species. For example:

GREEN 11th-13th

YELLOW 13th -16th

RED 16th -18th

TRIANGLES: HUMAN

CIRCLE: HERBIVORES etc.

- Fig. 5 Caption – The authors might want to consider inserting δ13C and δ15N (e.g., Plots of δ13C and δ15N values of herbivores, omnivores, carnivores, birds, and fish from the three countries (Belarus, Lithuania, Poland).

- Fig. 5 Legend – The authors might want to consider inserting different colours for different countries.

- Fig. 6 Caption – The authors might want to consider inserting δ13C and δ15N (e.g., Plot of δ13C and δ15N of human samples from sites in Belarus, Lithuania and Poland).

- Fig. 6 Legend – The authors might want to consider inserting the same colour and different shapes for different sites from the same country. Also, the authors may want to specify in what country are the specific sites in the legend to make it more clear to the readers: e.g., Kałdus (Poland), Alytus (Lithuania).

- As much as possible, the authors should consider using an aspect ratio close to 1 for graphs. In the authors’ version there is an apparent overestimate of the δ13C variability.

TABLES:

- Supplementary Table 1 – Number of adolescents is unclear.

REFERENCES

- Line 1051 = Brothwell DR. Digging up bones : the excavation, treatment and study of human skeletal remains. 3rd ed.. British Museum; 1981.

- Reference n. 84 is repeated twice [Line 1083 main text – Line 757 SI].

Reviewer #2: I welcome this paper which is concerned with medieval Belarus, a region of Europe which, as is rightfully made the point by the authors, has not just been understudied but has also been (albeit unconsciously) somewhat dismissed by bioarchaeological research. The article makes a valuable contribution, not just on account of the large new C&N isotope dataset it offers but also because the study is well contextualised with literature from the region which is otherwise not easily accessible to many researchers.

I am therefore very much in favour of the paper being published and believe methods and data interpretation are overall sound, although in several points, this is not easy for me to validate, because of the confusing way that the paper (and data) is presented. Some of this appears to be simply “lost in translation” and should be easily rectified by a thorough edit by one of the native English speakers among the authors. Other issues require a little more work. In general, I would encourage a bold restructuring of this paper, with much greater focus on clarity and brevity and moving much more of the overly detailed information into the SI.

For ease of reference, I have made a large number of comments and suggestions directly in the pdf. I believe most of the Discussion is sound but findings should be revalidated once data analysis (including Figures) have been revised.

As a general point, most of the Figures/data-plots are far too full and confusing at this moment – and this is before they have been reduced in size to fit on an A4 page. I have several suggestions in the pdf, but figures need attention in general.

6. PLOS authors have the option to publish the peer review history of their article (what does this mean?). If published, this will include your full peer review and any attached files.

Reviewer #1: No

Reviewer #2: No

---

## [Author Response · Author response to Decision Letter 0]

15 Aug 2022

We would like to thank the Editor and the two Reviewers for their constructive and useful comments on our manuscript. We are glad that they all considered the paper and the dataset to be an interesting and valuable contribution. We have tried to address all of the comments provided by the Editor and the two Reviewers, and we believe that we have now significantly improved the manuscript as a result. Our point-by-point responses in each case can be found below.

N.B. Where we refer to Line and Page numbers in our responses, these reference the previously submitted version of the manuscript and are the same as originally quoted by the Reviewers.

Editor

The archaeological record of the excavated sites is lengthy reported in the SI, but I urge the authors to provide some figures of the excavated graves and/or excavation plans. Unfortunately, almost all the archaeological reports are in Belarus language and difficult for the common reader to understand the differences between the funerary customs (e.g. kurgan vs other grave types) and the archaeological settings. Moreover more details should be provided about how the site chronology were determined.

We thank the Editor for this suggestion for improvement. To address this comment as well as the request by Reviewer 1 to better describe the various sites’ funerary contexts, we have added:

-the number of excavated individuals to Table 1 and the description of how this number was counted for the purposes of this article to the Supplementary Note 3 (new section “Number of graves and individuals”)

-expanded the Supplementary Note 3 with one more section “Burial custom: kurgan (mound) and ground (earth/pit burials)”, describing the funerary custom and its evolution and how the chronology of the sites was established

-figures with examples of kurgan and ‘stone’ grave plans, and several more cemetery excavation plans of the mentioned sites. Adding plans for all of the sites is not deemed possible, as most of them are available only in bad quality. Furthermore, almost all cover not the whole cemetery, but every layer and trench excavated in a certain year while, in the case of kurgans, there are very high-level "situational plans" and detailed plans of each excavated kurgan available. This would mean that close to a hundred images would need to be inserted.

We have therefore rather sought to provide some good case study examples. We hope now with this additional information that the archaeological context of the sites is clearer.

Bioarchaeology of the human remains, sex and age at death methods should be described and justified better. It is hard to believe to sex estimates made on the cranium only of age at death based on cranial sutures analysis. Moreover, the authors write “In the cases where sex and age estimation based on the study of the cranium contradicted those stated in the available publication or archaeological report, the latter assessments were used based on the assumption that it could have been estimated with more certainty at the time of analysis when postcranial elements were also available”. So, why did the authors a new anthropological analysis of the human remains?

We agree entirely with the Editor on this point, and have now tried to better address the concern with the credibility of bioarchaeological assessment. Unfortunately, a lack of postcranial elements in the case of the human specimens from one of the collections (National Academy of Sciences of Belarus) makes it impossible to assess the sex with greater precision. These, however, represent 35 out of 143 individuals. The decision to favour the sex and age estimation from the available publication or archaeological report was made specifically for this reason, as from the photos or the description of the remains (extended position of the skeleton, positions of hands and legs) in some reports it was clear that almost a full skeleton was preserved at the time of excavation, while the history of the collection includes an unfortunate loss of many elements and the preferential preservation of skulls by the anthropologists who curated it. However, the sex or age were not always reported for the remains, therefore a new anthropological analysis of the sample was deemed necessary, alongside the contradiction between the assessment that arose from this analysis compared to previous observations in some cases (3 with male/female estimated differently based on skull versus publication; 7 more cases where sex was estimated as “indeterminate” based on skull, but more specific assessment from the publication was adopted).

To make these reasons clearer, we have changed the corresponding text to:

“Sex assessment was based on cranial morphology [85]. In the cases where sex determination based on the study of the cranium contradicted that stated in the available publication or archaeological report, the latter assessments were used as the sex could have been estimated with more certainty at the time of excavation when postcranial elements were also available. Given that sex assessment based on cranium is less relaible, we conduct an additional check before interpreting the results of male to female comparisons.”

To further address the concern with the reliability of sex assessment based on cranium, we have conducted additional statistical tests and have added the following lines to the methodology (above) and results and discussion (below):

“Males (n = 67) and females (n = 61) present mean δ13C values of −22.1‰ ± 1.1 and −22.0 ± 1.0‰ and mean δ15N values of 10.3‰ ± 1.4 and 10.2‰ ± 1.6, respectively (Table 4 and Fig. 4). The t-tests comparing sexes showed no evidence for statistically significant differences for either δ15N (t = 1.046; df = 126; p = 0.297) or δ13C (t = 0.799; df = 126; p = 0.426); this result remains true if individuals with postcranial elements (male n = 53; female n = 40) are tested separately (δ13C: t = 1.212; df = 91; p = 0.229; δ15N: t = 0.439; df = 91; p = 0.662). Likewise, no differences are found between males and females buried in urban or rural cemeteries, or between sexes compared in each period separately.“

“Given that no differences were observed between the isotope values in the bone and dentin collagen samples, and the results of male to female comparison stay the same for both the entire sample and for individuals with postcranial elements when they are tested separately, we assume that neither the heterogeneity of samples nor the variability of methods applied for sexing (based on cranium or postcranial elements) have substantially affected the interpretation of the results.”

We also completely agree that the narrowly-reported age-at-death based on cranial sutures analysis may be misleading and unjustified. In this case where such a method was applied (the collection of Polack State University) a new anthropological analysis was impossible due to time constraints and the agreements with curators. Therefore we could only rely on the estimates provided by the local anthropologists and the methods they use. Recognizing the low reliability of the aging method based on cranial sutures, we have not attempted any age-related comparisons or analysis in the study, using the age estimation solely to establish that the individuals in the sample were predominantly adults. To address the rightful concern of the Editor and Reviewer 1, we have replaced the narrow age ranges in the “age” column of the Supplementary Table 1 with the age categories “adult” and “adolescent” to avoid providing unjustified information. We have also added a few further cautionary lines to the main text as follows:

“Also due to the necessity to rely on the prior anthropological analysis in the case of individuals from Polack State University osteological collection and recognizing the low reliability of the aging method based on cranial sutures, we report the age-at-death only at the level of age categories “adult” and “adolescent” in Table S1 and do not undertake any age-related analysis in the study, using the age estimation solely to establish that the individuals in the sample were predominantly adults.”

Reviewer 1

The paper “Medieval and Early Modern diets in the Polack region of Belarus: A stable isotope perspective” explores patterns of paleodiet of inhabitants of the Polack region – Belarus – across time (11th-18th centuries), between sex, settlement type (rural vs urban) and burial context (elite vs military) through carbon (δ13C) and nitrogen (δ15N) isotope analysis of human and faunal bone and dentine collagen. With its significant set of data of human (n=143) and faunal (n=105) remains from 15 rural graveyards and the town of Polack, the paper constitutes a valuable contribution to reconstructing dietary habits across significant historical periods in Belarus - an under-investigated region compared to other parts of Europe for the reconstruction of past human diets. I find particularly noteworthy the extensive archaeological and historical background given in the Supplementary Information and the integration of various archaeological, literary, and historical sources with δ13C and δ15N outcomes in the discussion. The paper also compares the data with δ13C and δ15N values from neighbouring territories, showing the main differences. The data are exhaustively reported in the Supplementary Tables (Supplementary Tables 1 and 2) and summarised for chronological periods, type of site and type of burials (Tables 3 and 4). Quality control is in place and reported in the Supplementary Tables. The statistical analysis performed is adequate for the study. The large chronological span and diverse type of site (rural vs city) provide the dateset for answering the research questions the authors outline.

We thank Reviewer 1 for this positive assessment of the significance of our dataset and our thoroughness of our analysis. We are also glad that they found the attempt made to e document the local sources related to the past diet and nutrition to the English-speaking audience to be important.

However, the manuscript needs some revisions. As a general comment, there are some typos and problems with consistency in how the results are reported. The authors might want to consider making some changes to the Figures.

Thank you for noting this, we address such issues below under the corresponding detailed comments.

All historical and archaeological bibliography is in Belarusian or Polish language, and it would be useful to have more general references in English.

Indeed, this is a trade-off between providing adequate coverage of the local context and the importance of accessibility for a wider audience. As Reviewer 2 notes, the literature from the region is not easily accessible to many researchers, therefore one of the aims and focuses of our manuscript was to make the local publications and archaeological reports available for the wider audience, who cannot otherwise access them due to language barrier, lack of digitised copies, and local regulations related to issuing reports, which require obtaining written permissions from the Academy and the living authors. In fact, beyond the isotopic dataset, this is a major contribution of our manuscript to the field. Furthermore, information related to the paleodiet in Belarus has not been found by the authors to be widely discussed in more internationally available resources or in English, therefore we relate to the studies from Poland and Lithuania published in English where possible (e.g. Dembińska and Weaver, 1999; Reitsema et al., 2010; Reitsema et al., 2017; Piličiauskienė and Blaževičius, 2019; Whitmore et al., 2019; Skipitytė et al., 2020). We have made another attempt to review the English literature related to palaeodiet in Belarus, but have not found more h sources that are easily available.

The various sites’ funerary contexts should be better described. The authors might want to clarify how reliable the given chronology is and explain how this latter was defined (e.g., based on archaeology/typology).

We thank Reviewer 1 for this helpful and important observation. We have made the changes as described above in the reply to the first comment of the Editor and hope they also satisfy the propositions of Reviewer 1 now.

A major concern is related to the age-at-death determination of adult individuals, which shows very narrow ranges compared to the methodology used. 

This is indeed a very true observation and, as noted in the reply to the 2nd comment from the Editor above, we found it most appropriate to replace those narrow estimations with broad age categories, leaving the age-at-death assessment at the level “adult” to “adolescent”.

Additionally, in the methods, the authors rightly highlight the difficulties of comparing bones with tooth roots as they reflect different years of life, but then this aspect is never taken into account in the discussion of results.

We have now added the following lines to the Human results section and Discussion:

“No differences were observed between the bone and dentin collagen samples for either δ15N (Mann-Whitney U test, U = 1778; p = 0,102) or δ13C (Student’s two-sample t-test, t = 0.153; df = 141; p = 0.879).”

“Given that no differences were observed between the isotope values in the bone and dentin collagen samples, and the results of male to female comparison stay the same for both the entire sample and for individuals with postcranial elements when they are tested separately, we assume that neither the heterogeneity of samples nor the variability of methods applied for sexing (based on cranium or postcranial elements) have substantially affected the interpretation of the results.”

Finally, some parts of the text in the discussion are slightly hard to follow and forced when integrating δ13C and δ15N analysis and historical data.

Assuming that those difficulties are listed in the detailed comments below, we hope they are now addressed in full as can be seen from the replies to follow. We have also taken another go through the Discussion more generally and tried to make the statements and comparisons clearer and more straightforward.

It follows some minor and major points that I suggest the authors amend:

MAIN TEXT:

MINOR:

- Indentation does not seem consistent across the main text and Supplementary Information [e.g., Lines 103, 251, 299, 314]. The authors might want to consider this aspect throughout the text.

Thank you for pointing this out, we have gone through the main text and the Supplementary Information and fixed the indentation.

- The spacing when reporting results and samples’ numbers does not appear to be consistent across the text. For example: (n = 49) [Line 401], (n=10) [Line 421], (n=51) [Line 439], (n = 142) [Line 463]; (t=1.894; df=141; p=0.06) [Line 472], (F = 6.381; df = 2; p =0.003) [Lines 438-439], (2.255; df = 2; p = 0.324) [Line 444]. The authors might want to consider this aspect throughout the text.

Thank you for pointing this out, we have now added spaces before and after the mathematical symbols (=, <, >, ±) in all cases for the sake of consistency.

- E.g. is reported sometimes with a comma, other times without a comma. For example: (e.g., maize and millet) [Line 188] vs (e.g. meat, aquatic) [Lines 195-196].

Thank you for pointing this out, we have now removed the comma after “e.g.” so the reporting is consistent in all cases.

- Lines 89-91 = The authors might want to give a range for the diverse chronology – “Ancient Rus principalities (9th century AD), the Grand Duchy of Lithuania (13th century ASD) and the Polish-Lithuanian Commonwealth (16th century AD)”.

Added ranges as suggested by both Reviewers, so the text now is:

“The time span of the sites covers the period from the 11th till the 18th centuries AD, and is subdivided into three roughly equal periods corresponding to periods of major state formation within the territory - Ancient Rus principalities (9-12th century AD), the Grand Duchy of Lithuania (13-16th century AD) and the Polish-Lithuanian Commonwealth (16-18th century AD).”

- Line 90 = Typo – “Lithuania (13th century ASD)”.

Thank you for pointing this out, we have fixed it now.

- Line 93 = Commoners vs elite – The definition appears a bit simplistic since nothing prohibits that there were individuals with greater status or wealthier among the commoners and vice versa.

This is a fair comment, and indeed there are a variety of ways one can define elites. In essence we compare groups in our sample which were presumed to possess a certain status (nobility) or occupation (military) by the archaeologists based on their context of burial to the individuals from rural graveyards whose burial context was not singled out as pointing to their status being different as a group. We find it to be an interesting outcome that no differences in stable isotope values were observed between the presumed elites while expected differences existed in the military subsample, and this information might add further value to the interpretation of the position of this group in its turn. To better reflect the above nuances, and because we do statistically compare the ‘elite’ and ‘military’ subsamples only to contemporary rural samples, we have made several changes to the text where it refers to ‘commoners’, and changed this particular line to: “Our goal is to determine whether there were temporal dietary changes brought by the associated political, economic, and social transition, both in urban and rural settings, in male and female individuals, and between the groups defined by archaeologists as nobility and military based on their burial context versus the rural individuals. We note that this division is fairly simplistic and may not always accurately reflect lived experience or status but, nevertheless, we believe that some broad insights into social and political elements of diet in past Belarus can be developed in this regard.”

- Lines 112-116 = Add reference or refer to SI – “Over the course of the period between the 11th and 18th centuries, these formations included Ancient Rus, the Grand Duchy of Lithuania (GDL), and the Polish-Lithuanian Commonwealth (PLC). The latter state ceased to exist at end of the 18th century and the Polack region, as well as other Belarusian lands, was gradually annexed by the Russian Empire”.

To address the comment of Reviewer 2 we have added more details about the state formations from the SI now, including the corresponding references, which hopefully addresses the comment of Reviewer 1 as well. The updated text now reads:

“The Polack Principality was the most prominent of the proto-states on the Belarusian lands, having its own ruling dynasty and maintaining certain independence even when it became part of larger territorial formations in this part of Eurasia [9,20]. Over the course of the period between the 11th and 18th centuries, these formations included Ancient Rus, the Grand Duchy of Lithuania (GDL), and the Polish-Lithuanian Commonwealth (PLC). Being one of the multiple states that together comprised Ancient Rus, the Polack Principality was involved both in prospering trade and numerous military campaigns and territorial changes [13,21]. Against the backdrop of battles with Tatar Mongols and the Teutonic Order a new state formed in Eastern Europe in the middle of the 13th century - the Grand Duchy of Lithuania, and from the 14th century AD, the lands of Polack became part of it. In the 15-16th centuries the culture of GDL was impacted by the European Renaissance; cities, craft and education flourished [22]. In 1569 the GDL and Poland united into the Polish-Lithuanian Commonwealth, and the following centuries were marked by devastating wars, depopulation and decline of the role of Polack as a political and socio-economic centre [9]. The PLC ceased to exist at the end of the 18th century when it was partitioned by Austria, Russia and Prussia, and the Polack region, as well as other Belarusian lands, was gradually annexed by the Russian Empire (refer to Supplementary Note 1 for more details on the historical background).”

- Lines 114-115 = “at end of the 18th century” – at the end.

Thank you for pointing this out, we have fixed it now.

- Lines 116-118 = Add reference – “As social identity is believed to be deeply embedded in diet and consumption behaviours, these events might have been paralleled by dietary changes”.

Added.

- Line 181 = “Stable isotope analysis and palaeodiet in Medieval eastern Europe” but Poland dataset includes “2nd century AD to 10-18th centuries”.

We agree that this reference was inappropriate for the stated topic of the section, and have now changed the line to: “For example, a series of isotope studies published by Reitsema, Kozłowski and others [2,3] investigated palaeodiet in Poland for the period ranging from Medieval Poland of the 10-14th centuries’ to the Polish-Lithuanian Commonwealth of the 16-18th centuries.”

- Line 232 = “Kurgan” – the authors might want to define the term.

Following the remarks of both Reviewers we have changed this line to “The human remains analysed in this study originate from burial sites that include kurgan (burial mound) and ground (pit burial) cemeteries in rural and urban environments.”

- Line 316 = The authors might want to add what types of other bone tissues were sampled as reported in Supplementary Table 1.

Added: “Human bone collagen was sampled primarily from rib bones (92/100), and, in several cases where ribs were lacking, from femur, fibula, ulna, and radius bones.“

- Line 397 = “Abandoned” – The authors might want to find another word.

We have changed the line to: “The animal sample which had a yield below 1% (Bir13-N2.01) was excluded from the analysis even though its C:N ratio was within the acceptable range of 2.9-3.6.”

- Line 435 = “Insignificant” – The authors might want to find another word.

We have changed to “not significant”.

- Line 448 = “Human bone collagen” – Human bone and dentine collagen.

Thank you for pointing this out, we have fixed the line as suggested.

- Line 463 = Typo – “Human (n = 142) mean δ13C values” – (n = 143).

Thank you for pointing this out, we have changed to “(n = 143)”.

- Line 531 = Add reference.

We have now added references to the ethnographic sources mentioned earlier in the text.

- Line 550 = Add reference.

We have now added references to the studies discussing the fasting and fish consumption mentioned earlier in the text.

- Lines 592-593 = “Polack inhabitants have δ15N values on average 0.5‰ higher than villagers”– should be 0.4‰ (rural δ15N Mean: 10.0‰; urban δ15N Mean: 10.4‰).

Thank you for pointing this out, we have fixed the mistake.

- Lines 661-662 = “As can be seen from Figure 3” – maybe Figure 5.

Thank you for pointing this out, we have changed to “Figure 5”.

- Line 759 = “As is also seen” – As it is also seen.

Thank you for pointing this out, we have fixed the line as suggested.

MAJOR:

- Lines 339-340 = In Supplementary Table I, age-at-death is reported with very narrow ranges (e.g., 55-60 or 45-50) mainly if this was estimated based on the cranial suture closure observation only – “The age-at-death was assessed observing cranial suture closure following Piontek [71]”.

We agree with that issue raised by Reviewer 1 and the Editor and address it in the comment to the Editor above. 

- Lines 344-345 = When estimating age-at-death based on dental wear patterns, the authors might want to use (Lovejoy 1985) instead of (Brothwell 1981).

We would prefer to stick to Brothwell (1981) as the decision in favour of this method was based on the fact that Brothwell's sample consisted of prehistoric to medieval English skeletons while the Lovejoy method is based on a North American population with hunter-gatherer diets. Therefore, the former is more population specific for the medieval and early modern European material we are dealing with in this study. At this present stage it is also not easily possible to return to the sampled remains to re-do the analysis. We do however now note that more than one method is available for this:

“In the collection of the Institute of History of the National Academy of Sciences, postcranial skeletal elements were not available. Age was estimated using dental wear patterns [88] coupled with analysis of the sphenooccipital synchondrosis fusion [89]. Our decision in favour of the age-at-death estimation method by Brothwell [88] was based on the fact that Brothwell's sample consisted of prehistoric to Medieval English skeletons while the later Lovejoy [90] method is based on a North American population with hunter-gatherer diets. Therefore, the former is more population specific for the Medieval and Early Modern European material we are dealing with in this study.”

- Lines 531-534 = The authors might want to reconsider this sentence. I am not entirely sure we can relate 19th-20th century ethnographic sources with what happened in the 11th-13th and 13th-16th century.

We are not sure what the Reviewer means in this case. The sentence in question is “Given that the picture of meat appearing on the peasants’ table on festive occasions was drawn primarily from ethnographic sources [21,26,27], which relate to the 19-20th centuries, this might suggest that animal protein before that time was actually more common in the diet of commoners“. Therefore, it is meant to contrast the 19-20th century available data to the earlier situation, not to use one to prove or disprove the other. We have now added a few further clarifications here:

“Given that the classic picture of meat appearing on the peasants’ table on festive occasions was drawn primarily from ethnographic sources [26,31,32], which only relate to the 19-20th centuries, our isotopic data might suggest that animal protein before that time was actually more common in the diet of commoners“

- Lines 502-507 = The sentence is slightly unclear and imprecise. Additionally, the authors might want to explain better what type of omnivores were considered.

We are grateful to the Reviewer for raising this concern. For greater clarity and precision we have changed the sentence to ‘A comparison of domestic omnivore (pig) δ13C and δ15N between sites and chronological periods indicates no statistically significant differences, while the domestic herbivore sub-sample showed a slight, statistically significant increase in δ13C between the groups of sites dated to the 11-13th centuries AD and the other two periods.”

- Lines 590-614 = The interpretation is slightly unclear – maybe the authors want to consider reframing this part and contextualising it more clearly. I also think that the sentence “Considering that this is a more expensive source of energy, it might mirror the important economic role of Polack in reflecting the generally higher living standards and food security of its inhabitants” might be an overstatement since 0.4‰ (general urban vs rural) or 0.5‰ (13th-16th century) is a too little shift.

Text: “In terms of the comparison between city and village populations, we can see that urban or rural settings did not have too much of a major role in determining diet, at least at the resolution of isotopic analysis. Polack inhabitants have δ15N values on average 0.5‰ higher than villagers. This small difference is significant only in the 13-16th centuries; however it is notable that at least the urban herbivores at this time have lower δ15N values. A possible explanation is a slightly higher proportion of animal or freshwater fish protein in the urban diet, which could have been more available in such an important economic centre as Polack. The reason why the difference is visible only between the 13th and 16th centuries may be due to the fact that prior to this point the differentiation between the town and countryside was yet emerging, with the majority of city dwellers engaged in agriculture and husbandry [9]. The lack of significant difference between rural and urban samples in the 16th - 18th centuries may reflect the econimic decline of Polack due to constant military conflicts and raids on northern Belarus by Russian troops that became regular since the 1510s, and in particular the Livonian War of the second half of the 16th century - all of which resulted in the influx of rural population [9], potentially contributing to the observed similarity between the diests of the inhabitants of Polack and the surrounding countryside. Despite the initial proposition that urban diets could be more heterogeneous due to obtaining food from markets where not only strictly local but more distant and even foreign foods could be available, urban dwellers seem to, overall, cluster rather tightly, in a similar manner to the rural communities.”

- Lines 623-647 = When discussing “what factors could have caused this shift” in δ13C and δ15N values after the 16th century, there is a bit of overreaching when explaining the possible increase in millet consumption. The authors might want to reconsider this aspect.

We agree and have now reduced this portion of the discussion to leave it more equivocal as follows: 

“However, as the change seen following the 16th century cannot be explained by other parameters such as the environmental baseline (it showed a different pattern) or the variable nature of the samples dated to various periods (a similar pattern is preserved if rural and urban or male and female groups of individuals are compared separately by period), it is important to discuss what factors could have caused this shift. If the δ15N values are simplistically interpreted as greater access to a higher trophic or aquatic protein source, then the slight increase of them in the time of the Polish-Lithuanian Commonwealth would be in line with the picture of the “Golden Age”, not only for the highest strata of the society but also more broadly across the population. The simultaneous increase in δ13C values in the period following the formation of the Polish-Lithuanian Commonwealth may also reflect the adoption of 13C-enriched fish into the diet by part of the population, which would be consistent both with the reported adoption of some marine imports into the menu of the nobility and with the striking reliance on such fish reported by Reitsema et al. [2] among Polish nobility of the same time period. Finally, the increase in human δ13C values could indicate higher millet consumption in this period. This would perhaps fit with increased reliance on millet during the adverse weather events of 1600-1603 and the influence of the Little Ice Age, and the aforementioned constant military conflicts starting from the early 16th century, which destroyed harvests among other things and undermined the economic contributions of the territory [34,100]. That said, the corresponding change in δ15N would perhaps imply that aquatic resources are the more likely contributors. Overall, human diets remain remarkably consistent between social groups and between time periods from an isotopic perspective.”

FIGURES:

- Fig. 1 Legend – The authors might want to consider adding in the legend the dates in the format reported in the text (e.g., 11-13th AD). The authors might want to consider spacing the legend animal-period from human-period a bit more.

We have applied the recommended changes reporting dates as “11-13th c. AD” and increasing the spacing in the legend.

- Fig. 3 Caption – The authors might want to consider inserting δ13C and δ15N (e.g., Plot of δ13C and δ15N values of human and animal samples compared by period).

Thank you for the proposition, we have changed the caption to the line suggested by the Reviewer.

- Fig. 3 Legend – The figure appears slightly hard to read – the authors might want to consider giving the same colour for each period and the same shape for each species. For example:

GREEN 11th-13th

YELLOW 13th -16th

RED 16th -18th

TRIANGLES: HUMAN

CIRCLE: HERBIVORES etc.

According to the comments by both reviewers we have now significantly simplified this figure by changing individual data points for animals to mean values and splitting them by period only where it would make a meaningful comparison (where the sample dated to each period would be >=5). We have also given the same colour to each period and shape to each species as recommended by Reviewer 1.

- Fig. 5 Caption – The authors might want to consider inserting δ13C and δ15N (e.g., Plots of δ13C and δ15N values of herbivores, omnivores, carnivores, birds, and fish from the three countries (Belarus, Lithuania, Poland).

Thank you for the proposition, we have changed the caption to the line suggested by the Reviewer.

- Fig. 5 Legend – The authors might want to consider inserting different colours for different countries.

We have now inserted different colours for different countries.

- Fig. 6 Caption – The authors might want to consider inserting δ13C and δ15N (e.g., Plot of δ13C and δ15N of human samples from sites in Belarus, Lithuania and Poland).

Thank you for the proposition, we have changed the caption to the line suggested by the Reviewer.

- Fig. 6 Legend – The authors might want to consider inserting the same colour and different shapes for different sites from the same country. Also, the authors may want to specify in what country are the specific sites in the legend to make it more clear to the readers: e.g., Kałdus (Poland), Alytus (Lithuania).

We have now given the same colour to each country and different shapes to different sites within it, and added the countries to the legend.

- As much as possible, the authors should consider using an aspect ratio close to 1 for graphs. In the authors’ version there is an apparent overestimate of the δ13C variability.

This is indeed a fair comment and we thank Reviewer 1 for pointing out this problem. We have now changed the aspect ratio in all figures to 1.

TABLES:

- Supplementary Table 1 – Number of adolescents is unclear.

We have now changed the column with age ranges in this table to contain only the broad definitions “adult” or “adolescent” following the remarks regarding the unjustifiably narrow age ranges. This should hopefully also make the number of adolescents clear in this table.

REFERENCES

- Line 1051 = Brothwell DR. Digging up bones : the excavation, treatment and study of human skeletal remains. 3rd ed.. British Museum; 1981.

We are not sure what the Reviewer meant by this comment, unless it referred to the double dot after “3rd ed”, which we now deleted.

- Reference n. 84 is repeated twice [Line 1083 main text – Line 757 SI].

Thank you for pointing it out, we have fixed the numbering in Supplementary Information. Now it starts from 106 due to the overall changes to the text and addition of some references.

Reviewer 2

I welcome this paper which is concerned with medieval Belarus, a region of Europe which, as is rightfully made the point by the authors, has not just been understudied but has also been (albeit unconsciously) somewhat dismissed by bioarchaeological research. The article makes a valuable contribution, not just on account of the large new C&N isotope dataset it offers but also because the study is well contextualised with literature from the region which is otherwise not easily accessible to many researchers.

We would like to thank Reviewer 2 for their highly positive comments on our manuscript. We are glad they find our data to be valuable and also that they find the compilation of the literature from the region to be important.

I am therefore very much in favour of the paper being published and believe methods and data interpretation are overall sound, although in several points, this is not easy for me to validate, because of the confusing way that the paper (and data) is presented. Some of this appears to be simply “lost in translation” and should be easily rectified by a thorough edit by one of the native English speakers among the authors. Other issues require a little more work. In general, I would encourage a bold restructuring of this paper, with much greater focus on clarity and brevity and moving much more of the overly detailed information into the SI.

For ease of reference, I have made a large number of comments and suggestions directly in the pdf. I believe most of the Discussion is sound but findings should be revalidated once data analysis (including Figures) have been revised.

As a general point, most of the Figures/data-plots are far too full and confusing at this moment – and this is before they have been reduced in size to fit on an A4 page. I have several suggestions in the pdf, but figures need attention in general.

We thank the Reviewer for all of these comments. In what follows, we hope to show that we have acted on all of their suggestions and greatly appreciate them.

Suggestions from pdf:

- Line 49-50 = Meanwhile?

We are not sure what the Reviewer meant by this comment, but we assume that the choice of the word “meanwhile” was considered inappropriate, so we have now changed it to “in particular”.

"The Medieval period of Europe witnessed a series of major economic, political, and social changes, including the rise of Christian states, the expansion of urban networks together with facilitated disease transmission, the intensification of hierarchies and wealth inequality, and increased access to non-local resources [1–4]. While these broader processes are well-documented historically and archaeologically, their impacts on local populations in different parts of Europe are often more obscure, particularly given that archival records are often biased towards elite communities [1,4,5]. In particular, eastern Europe, located beyond the geographic scope of Roman Christianization, has also received less attention and is often considered to be poorer, less developed, and less “European” [5]."

- Line 59 = ethnicities

Changed “ethnoses” to “ethnicities” as suggested

- Line 59-61 = clarify: do you mean in relation to bioarchaeology or are you suggesting that noone, medieval historians included have ever looked at any of these issues?

Thank you for pointing this out, we have changed the text to clarify as follows:

“Being part of the Grand Duchy of Lithuania, at its height the largest state in Medieval Europe, and described in the literature as a melting pot of ethnicities, religions and cultures [11], the region thus remains a yet untapped opportunity to explore issues pertaining to identity, equality, human agency, resilience and adaptation to socio-political transitions in the Medieval world through the use of the state-of-the-art methods offered by bioarchaeology.”

- Line 67 = define in terms of centuries for your area. I would not have thought you cover the early medieval period, but boundaries may well be different in your region

We have added the explanation of the periodization we use as follows:

“The time span of the sites covers the period from the 11th till the 18th centuries AD, and is subdivided into three roughly equal periods corresponding to periods of major state formation within the territory - Ancient Rus principalities (9-12th century AD), the Grand Duchy of Lithuania (13-16th century AD) and the Polish-Lithuanian Commonwealth (16-18th century AD). We refer to them as High, Late Medieval and Early Modern periods following the general division accepted in European history here, though we admit the local variation of this periodization. ”

However in this case “Early Medieval” was taken from the reference (where it referred to the 9-10th centuries). Admitting that the periodization may vary not only within Eastern Europe, but within one country (e.g. in Belarus there is a debate how to define the local periodization of the Middle Ages and the propositions vary greatly in terms of centuries to which they refer), we have decided to avoid engaging in this discussion and following this comment we have changed the line in question as follows:

“While yielding high levels of detail on cuisine and foodstuffs consumed, written records are still virtually nonexistent for the first centuries of the 1st millennium in certain parts of eastern Europe [9,12,13] and, where available, historical documentation revolves around social and religious elite groups [5].”

- Line 70 = Revise: this depends entirely on the time it took to accumulate the context in question, doesn't it? More often, archaeological contexts relate to extended periods rather than 'moments'

We agree with the reviewer that this phrasing is misleading. To better reflect the nature of zooarchaeological and archaeobotanical limitations as pointed out by Reviewer 2, we have changed the text according to this and the next comment: 

“Zooarchaeological and archaeobotanical data speak to the specific foodstuffs available for a given, context-based window of time but do not necessarily provide insights into the long-term dietary reliance of an individual [12]. They are also limited by taphonomic biases [14] and retrieval techniques, and are not always capable of showing how available resources were distributed within the population through time [3].“

- Line 72-73 = again, I disagree. You can contrast assemblages from different contexts (e.g. high and low status sites, urban vs rural). A lot can be done there.

Assemblages certainly provide information down to community (i.e. site) level 

We agree with Reviewer 2 that such possibilities exist, in this case we rather meant to give an example where the authors of a study found such a challenge, rather than to say that it is impossible to study how resources were distributed within the population through archaeological context, at least indirectly. Therefore, we have changed the wording as shown in the reply to the previous comment.

- Line 84 = is everyone buried in a rural graveyard a 'peasant' - perhaps define the term / explain in more detail in your methods section and use a more neutral term here.

Even if they are all peasants, this is a large and heterogeneous group in the Middle Ages

We thank the reviewer for this criticism and have now changed the term to more neutral “individuals from rural graveyards“, “rural dwellers/commoners/sample”, or “village population” throughout the text to address it.

- Line 86 = contemporaneous

Changed “contemporary” to “contemporaneous” as suggested

- Line 89-91 = clarify: This suggests that your 'periods' only cover one century each = and the first one even a century you do not have samples from.

I assume you mean 9th-12th, 13th to 15th and 16-18th centuries, but this needs to be clear 

You need to provide some context to explain these terms (see below)

Added periods as suggested by both Reviewers, so the text now is:

“The time span of the sites covers the period from the 11th till the 18th centuries AD, and is subdivided into three roughly equal periods, corresponding to periods of major state formation within the territory - Ancient Rus principalities (9-12th century AD), the Grand Duchy of Lithuania (13-16th century AD) and the Polish-Lithuanian Commonwealth (16-18th century AD).”

Further details about the context were added as suggested and replied in the below comments.

- Line 105-106 = explain what this phrase relates to and where it derives from

Changed the text to: Formed on the trans-European trade route “from the Varangians to the Greeks”, described in the Russian Primary Chronicle as connecting Scandinavia (Varangians) with Byzantium [18,19], the city of Polack (also Polatsk, Polotsk; official transliteration from the Belarusian language is used here for Belarusian toponyms and names, spelled with Belarusian Latin alphabet) was the major trade, political and cultural centre of the region [7,9].

- Line 109 = 

Added “The Polack Principality”

- Line 113-114 = Details on these are needed. I can see that you have a lot in the SI but this is not referred to here and also you need to give at least a very short overview of the main themes and socio-economic changes here. The article has to make sense for readers without referring to the SI.

Changed the text as follows:

“The Polack Principality was the most prominent of the proto-states on the Belarusian lands, having its own ruling dynasty and maintaining certain independence even when it became part of larger territorial formations in this part of Eurasia [9,20]. Over the course of the period between the 11th and 18th centuries, these formations included Ancient Rus, the Grand Duchy of Lithuania (GDL), and the Polish-Lithuanian Commonwealth (PLC). Being one of the multiple states that together comprised Ancient Rus, the Polack Principality was involved both in prospering trade and numerous military campaigns and territorial changes [13,21]. Against the backdrop of battles with Tatar Mongols and the Teutonic Order a new state formed in Eastern Europe in the middle of the 13th century - the Grand Duchy of Lithuania, and from the 14th century AD, the lands of Polack became part of it. In the 15-16th centuries the culture of GDL was impacted by the European Renaissance; cities, craft and education flourished [22]. In 1569 the GDL and Poland united into the Polish-Lithuanian Commonwealth, and the following centuries were marked by devastating wars, depopulation and decline of the role of Polack as a political and socio-economic centre [9]. The PLC ceased to exist at the end of the 18th century when it was partitioned by Austria, Russia and Prussia, and the Polack region, as well as other Belarusian lands, was gradually annexed by the Russian Empire (refer to Supplementary Note 1 for more details on the historical background).” 

- Line 126-127 = Unnecessarily complicated. Better?: Rye, followed by wheat, barley and oats were the most common cereals

Thank you for the suggestion, we have now changed the text from “Wheat, barley and oats followed rye in popularity [10,24,25,28], along with garden crops like cabbage and turnip [7,20,28-30].” to “Rye, followed by wheat, barley and oats were the most popular cereals [10,29,30,33], while garden crops like cabbage and turnip were also common [7,25,33–35].”

- Line 128-129 = Consider taphonomic reasons for this (millet tends to be underrepresented in charred assemblages), see Roesch M, Jacomet, S, Karg, S (1992) Veget Hist Archaeobot 1: 193 -231

Make point that this makes it even more important to investigate through isotope analysis.

Thank you for this idea for improvement, we have now added the consideration of taphonomic bias, the new reference, and the note on importance of isotope analysis. Now the text is as follows:

“By contrast, millets appear in rather negligible quantities among charred grain finds and continue to decrease in number over the period of our focus [29], though some authors name millets as a widely grown crop and even one of the main foods in Ancient Rus [7,33]. As millet was found to be underrepresented in charred grain assemblages [36], this may also be due to taphonomic reasons. In such a light, isotope analysis can provide a means to directly validate the extent of consumption of millets.”

- Line 166-167 = what do you mean by "marginal"? Something that isotope analysis would not pick up? Rephrase

As before: would avoid the term peasant

We have rephrased the sentence as follows: “Therefore, if differences based on access to rare foreign foods existed between rural and urban dwellers, they were likely rather negligible and more applicable to wealthy citizens than commoners.”

- Line 168-169 = Again, rephrase (no such thing as "abundant nutrition")

Also: nutrition is not the same as diet.

Changed “nutrition” to “diet”.

- Line 179 = End with an assessment of which distinctions you might be able to pick up in the isotope record

We have now added the following text at the end of this paragraph:

“Stable carbon and nitrogen isotope anlaysis may shed light on some aspects of the diet described above, such as the dominance of plant-based diets and the differing access to animal protein among various groups like elites, rural and urban dwellers. It has also proven useful in detecting the consumption of millets and fish, while potential enrichment in 15N of domestic species may support the use of manure on crops that they would then subsequently feed. Other changes in diet, such as increased incorporation of imported foods may be only observed if these foods were isotopically distinguishable - for example, marine products.”

- Line 224 = better: little evidence for

Thank you for the proposition, we have changed “low fish consumption” to “little evidence for fish consumption”.

- Line 232 = I expect this one is common knowledge,but still clarify what a kurgan is in brackets after the term.

Not a term I am familiar with. flat graves? simple earth graves?

Following the remarks of both Reviewers we have changed this line to “The human remains analysed in this study originate from burial sites that include kurgan (burial mound) and ground (pit burial) cemeteries in rural and urban environments.”

- Line 241 = ?? in what form?

Here we meant the crosses that Christians wear on the neck, and which are also rarely found in Christian burials along with the lack of other goods. Since the Reviewer found this line confusing, we have removed the mention of crosses as this information was not important for the study. The updated line sounds as follows:

“During most of the period of our focus, in the 14-18th centuries AD, the deceased were interred without any grave goods [52,54].“

- Line 241-246 = Give detail on the nature of the rural settlements that belong to the cemeteries. Refer to Table 1 here.

To address this and the below comment referring to Line 251-264, we have added a paragraph with clarification to the new “Burial custom: kurgan (mound) and ground (earth/pit burials)” section in the Supplementary Information:

“Unfortunately, only in the case of Polack can the described cemeteries be directly related to the settlement, while in case of the other sites no such data is available in publications. They are therefore defined as rural based on the lack of any known urban settlements in their vicinity, contrary to sites like Polack where the city still exists or is known to have previously existed based on archival or archaeological data (for example the ground cemetery near the village of Pašavičy, which is referred to as urban due to the archaeologically-documented early medieval city nearby [70]. Therefore, we describe only the rural cemetery sites without reference to rural settlements below.”

We have added a reference to Table 1, now the text is as follows:

“Rural cemeteries that provide the sample of village population in this study include Biruli, Domžarycy, Doŭhaje, Dubraŭka, Dzmitraŭščyna, Ivieś, Kazloŭcy, Klieščyno, Michalinava, Padsvillie, Pieravoz-4, Sielišča, Skrabianiec, Vaŭča and Ziabki (Table 1). Fig 1 shows a map of the sampled sites grouped into the three periods (11-13th, 13-16th, 16-18th centuries AD) of major state formation mentioned above (Ancient Rus, the GDL, the PLC).”

- Line 248-250 = Figure 1: I don’t expect this figure is good enough quality to be reproduced and would encourage for it to be redrawn, rather than just using a Google Maps screenshot with overlays.

As a minimum, however, use an English-language map and add some major landmarks for orientation (e.g. Minsk).

Revise legend to English (cc for ‘centuries AD’ is not any form I recognise). Use lines to connect labels with points on the map. Very difficult to understand what belongs where.

Change colours, so that the same colour indicate the same time scales – human and faunal samples are still indicated by different symbols. 

We have applied the recommended changes using a contour map, adding some landmarks, revising the legend, adding lines to connect labels with points on the map, and changing colours to be the same for the same period.

- Line 251-264 = Move detail into a 'site gazetteer' in the SI and report the most important info in Table 1.

More important here to give some general information about the nature and economy of the settlements in general and any change over time (or clarification that we lack this kind of information)

Since these details are present in the Supplementary Information and Table 1 contains the dating and type of cemetery, we have changed the paragraph in question to a single line as suggested: “The dating and cemetery types of each site along with the number of individuals excavated and sampled from them can be found in Table 1”.

Regarding the nature of the settlements see the previous reply to Line 241-246.

- Line 276 = add a column for site type: urban/rural, possibly high or low status?

cc AD is not an abbreviation I recognise.

We have now added a column with site type (urban/rural) and burial context (rather than status to accommodate the military sub-sample in it) to Table 1.

We have replaced the abbreviation “cc” for centuries with “c. AD" or "c." in all places.

Report individuals sampled next to how many individuals were identified in each cemetery in total, to give an idea of how representative your sampling is

We have now added such information to Table 1 (see reply to the first comment of the Editor).

- Line 289-290 = I don't know what this means. Rephrase to clarify

For clarity we have changed the line to: “The Township was the ancient centre of Polack, which was turned into a cemetery in the 17-18th centuries AD. This cemetery was excavated in 2007 and 2009 and is interpreted as being Catholic from the scarce inventory of burial goods uncovered.”

- Line 298 = Clarify that many of the faunal remains are not actually from the same sites as the human remains

To clarify this we have changed the paragraph as follows:

“Due to the fact that the exact isotopic ratios at the base of the food chain can vary across space and time as a product of various environmental and economic parameters it is necessary to sample contemporaneous fauna to fully understand and interpret the human results [15]. Part of the animal sample in this study originates from the same sites as human remains - the city of Polack (Lower Castle and Upper Castle) and the settlement site and one of the kurgans of Biruli. The rest of the animal remains come from other sites associated with the Polack Principality territory - Lučna (15-17th centuries AD) and Čarscviady (11-12th centuries AD), rural sites that hosted nobility residences, and the Mienka township (10-11th centuries AD) (Table 2). Fauna was identified by Dr. Urszula Iwaszczuk and Vera Haponava using the reference collection at the Institute of Mediterranean and Oriental Cultures of the Polish Academy of Sciences and a standard zoological reference work [81,82].”

- Line 310 = Give detail on species sampled or at least trophic category (e.g. wild/domesticated herbivores, omnivores, freshwater fish).

Species can then be in SI, if needed (I believe they are) - although would prefer them here for ease of reference.

We have now added a column with a list of species to this table.

- Line 313 = Clarify: What proportion of individuals from the sites were sampled?

Following the related comments from both Reviewers and the Editor we have now added the information about the number of individuals excavated in each site where it was available to Table 1, therefore we hope this comment is now also covered. We have also added the proportion of individuals sampled to this line:

“A total of 100 human bones and 43 human teeth were sampled in this study, representing 5-100%, on average 44%, of the (known) excavated individuals at each site (refer to Table 1 for the numbers and to Supplementary Note 3 for the explanation of how the numbers were counted).”

- Line 374-375 = one or two s.d.? (Here and for data above). You state below that you used 2 sd throughout but that seems unlikely here?

Thank you for this observation, indeed we report 1 standard deviation here, therefore now we have clarified it in the paragraph (and separately in other places where we use different std.dev.):

“Samples were combusted in a Thermo Scientific Flash 2000 Elemental Analyser coupled to a Thermo Delta V Advantage Mass Spectrometer at the Isotope Laboratory, Max Planck Institute for the Science of Human History (Jena, Germany). All isotopic measurements refer to the ratio between heavy and light isotope (13C/12C or 15N/14N) measured as δ values in parts per mil (‰) calibrated using a two-point calibration between a series of International Standards (IAEA-N-2 Ammonium Sulfate: δ15N = +20.3 ± 0.2‰, USGS40 L-Glutamic Acid: δ13C = −26.389 ± 0.042‰, δ15N = −4.5 ± 0.1‰, IAEA-CH-6 Sucrose: δ13C = −10.449 ± 0.03‰) and in-house laboratory standards (fish gelatin: δ13C = −15.7 ± 0.1‰, δ15N = −14.3 ± 0.1‰; Urea: δ13C = −41.30 ± 0.04‰, δ15N = −0.32 ± 0.2‰; ± 1 standard deviation is reported here). All samples were measured in duplicate. Analytical error was studied through the repeated measurement of an in-house fish gelatin standard (-15.7 ± 0.3‰ for δ13C and 14.3 ± 0.2‰ for δ15N; ± 1 std. dev.).”

- Line 377-381 = I would consider using non-parametric tests as a standard. There is no reason to expect isotope data to be normally distributed

We have considered this approach and though we agree that there is no reason to expect the isotope data to be normally distributed, however, we expect that not using non-parametric tests as a standard should not have had any negative effects on the quality of the analysis, especially given that they are generally considered less powerful than their parametric counterparts and are often used as a standard approach in similar studies based on stable isotopes (e.g. Riccomi et al., 2020; Hofman-Kamińska et al., 2018 PLOS ONE publication; Lightfoot et al., 2012; Reitsema et al., 2010). We have made no assumptions regarding the normal or not normal distribution, but checked if the distribution of data is close to normal in every sample group and applied the appropriate test accordingly.

- Line 388 = Perhaps in the text but not in the tables, as far as I can see? Better clarify each time you quote s.d.s

Indeed, as Reviewer 2 rightly noticed, the columns “SD” in the Tables 3 and 4 report 1 SD, while 2 SDs are reported throughout the text. To make it clearer we have changed the columns’ titles to “1 SD” and changed the line in question to “Means are reported ± 2 standard deviations in the below text, unless indicated otherwise (e.g. in figures and tables)“.

- Line 394 = Clarify presumably, you excluded below 1%?.

This information was included further in the same paragraph, but we have tried to make it clearer by changing it as follows:

“Collagen yields in the whole sample ranged from 0.002 to 26.36%. The animal sample which had a yield below 1% (Bir13-N2.01) was excluded from the analysis though its C:N ratio was within the acceptable range of 2.9-3.6. Three more animal samples were excluded from the analysis as their C:N ratio was outside of the acceptable range of 2.9-3.6 (Bir13-N5-6.02, Bir14-N1A.01, and PUC16-NV2.3.03). Therefore, the final number of samples used in the analyses was 244 out of 248 (Table S2).”

I would move the detailed discussion of what you excluded and why into SI and only report broad principles and numbers here.

Unless the Editor objects, we would prefer to keep this paragraph as it is rather short (9 lines) and it is not worth creating a new Supplementary Information section out of it, while including the information about the preservation parameters applied to samples and the criteria for exclusion is rather standard for similar papers.

- Line 394 = ?? This is more collagen than present in fresh bone?

Such collagen yield could be a preservation factor, such as increased demineralisation and preferential loss of inorganic component in some burial environments (e.g. Turner-Walker et al. 2006 experimentally showed that in humid and cold conditions collagen yield increases in bone) or a basic laboratory error when weighing the collagen (e.g. a bit of parafilm stuck on the tube when it was weighed). Given that collagen yield is dependent on such factors and all the other quality criteria, particularly C/N ratio are in range, we have not excluded such samples.

- Line 400 = No need to repeat all the statistics that are already summarised in Table 3. Can make better use of your words here.

I am concerned that you at least appear to simply pool groups of animals without prior testing whether there are differences between species or testing for differences between sites and regions. This is even though we might expect differences based on soil type and you told us earlier, that there are suggestions that manuring was more important in the south than the north, so you might expect differences.

Testing for regional variation is arguably more important than just testing for differences between pooled urban and rural groups, especially since animals found in urban deposits were more often than not raised in the countryside.

You can move some of these comparisons into the SI but need to make them.

P.S. Having read on, I see that you do make site comparisons below. I would move this up to not confuse readers and make sure that the summary statistics are included in Tables and that data from different sites are compared in a figure (Can be SI)

Thank you for bringing out this point - as a cumulative response to several comments from Reviewer 2 we have restructured this section, removing the descriptive statistics from the text and leaving them in the tables, adding the descriptive statistics by site to the Table 3 and description of how several of the sites were grouped to the table’s title. Now the initial description is limited to:

“The descriptive statistics (range, mean, standard deviation and median) for animal groups by feeding category (i.e. herbivore, omnivore), time period, species and site can be found in Table 3, while the plot of the animal δ13C and δ15N values is shown in Fig 2. Wild species in our sample included beaver, hare, bison, elk, red deer and wild bird. All excluding the unidentified wild bird are herbivorous animals (n = 8). Birds (n = 10) are mostly represented by chicken, but also geese and birds only defined to the level of ‘galliformes’ or wild bird. The fish (two pikes and one unidentified species) bone samples provide approximate δ13C and δ15N for available freshwater resources. Freshwater, anadromous and marine species from studies conducted in Poland and Lithuania serve to complete the picture of potential food sources in this study.”

This is followed by Table 3 and by the results of the statistical tests: inter-site and by period comparisons, and comparison between wild and domestic herbivores.

We have also added a new Figure S1 to the Supplementary Information, comparing animal samples from different sites and humans from rural and urban sites, with the following caption:

“Supplementary Fig 1. Plot of δ13C and δ15N values of human and animal samples (mean ± 1 std. dev.; individual values for groups where n < 6) grouped by sites. Due to the low number of animal samples from some sites, samples from Čarscviady were considered in the statistical tests and are shown here in a group with Lučna, and Polack Upper Castle together with Lower Castle. Human samples are shown in the two groups which are discussed in the text (all rural sites versus all urban/Polack sites).”

- Line 401 = Clarify: are there differences between species (cattle and sheep/goat?) or between different regions/sites?

Also per the comment above we have brought forward the information regarding inter-site comparisons, followed by comparison across time periods and wild/domestic species. We hope this is now all clearer.

We have also added a remark about the differences between species in results and discussion:

“As for the inter-species comparison, while a Kruskal–Wallis test again shows no differences between cattle, sheep/goat and horse/pony in δ15N values (H = 5.271; df = 2; p = 0.072; Table S8), the horse/pony group is 1‰ lower in δ13C than cattle and this difference is significant based on a Kruskal–Wallis test (H = 9.301; df = 2; p = 0.010; Table S7 for post-hoc tests).”

“As for the 1‰ difference in δ13C between horse/pony and cattle, similar lower δ13C values in horses in comparison to cattle and sheep/goats were found in other studies and interpreted as related to differences in pasturing strategies between species or as potentially caused by metabolic processes [94–96]. This difference may contribute to the higher δ13C values observed among domestic fauna in the 11-13th centuries, as the proportion of fauna represented by cattle is the biggest for this period (n = 18 compared to 8 in 13-16th centuries and 2 in the 16-18th centuries), while both horse and cattle samples are rather evenly distributed among sites.”

- Line 403 = it is important to tell us the species here, as they will all have different ecologies that may explain the isotope values

We have now added the list of species to the text: “Wild species in our sample included beaver, hare, bison, elk, red deer and wild bird. All excluding the unidentified wild bird are herbivorous animals (n = 8).”

- Line 408 = Preface statistics with which test you used here (and for other tests, too). I suggest you round t to a reasonable number of digits

Thank you for pointing out, we have rounded the t value in this line to 3 decimal points as everywhere else.

We have also satisfied the request regarding prefacing statistics with the name of the test in addition to quoting the test statistic in each case.

- Line 410 = leave interpretation to discussion

Deleted the interpretative part of the sentence “perhaps linked to the consumption of manured crops.”

- Line 414-415 = I expect this will be properly formated for the final submission? Adding gridlines would help.

Table should detail summary statistics by species, then, if you want, a pooled value.

Should also give statistics for different sites. Can move into SI, if needed.

We have fixed the format of the table (both Tables 3 and 4 and the Supplementary Tables 1 and 2) by adding gridlines and bold headers where they were lacking. We have also added the statistics for different sites in a way similar to periods and for species where n>=5 to Table 3.

- Line 417-424 = Move information into table

Done.

- Line 431-432 = which herbivores? Be clear what you are comparing

Added “domestic herbivores” everywhere in the paragraph.

- Line 432-433 = where are results of post-hoc tests?

We have now added these details to the paragraph as follows: 

“When faunal δ13C and δ15N across sites are compared, only differences in δ13C among domestic herbivores appear to be significant based on one-way ANOVA (F = 7.102; df = 3; p < 0.001). However, the differences appear to be ‘site-based’ and do not simply fall along a rural and urban division: in particular, post-hoc Bonferroni tests (Table S3) show that the Polack sites and rural sites with nobility residences Lučna and Čarscviady exhibit lower δ13C values than the rural site Biruli and the township Mienka (Table 3, Fig S6).”

We have also added post-hoc tests’ results to the Supplementary Information.

- Line 433-434 = indeed (see my comment above) - refer your reader to where they can see the descriptive statistics and graph for these differences.

We have now added the descriptive statistics by site to Table 3, a reference to it and to the plot with comparisons among sites in this line.

Also, again, be clear what you actually compared: presumably domestic herbivores only from the different sites?

We now refer to “domestic herbivores” across the whole paragraph, so this should hopefully be clear. E.g. the next line looks like: 

“At the same time, differences in the domestic herbivore δ15N values, not significant when compared with Kruskal–Wallis test between the groups of sites mentioned above (H = 6.052; df = 3; p = 0.109; Table S4), are significant at the level of the rural to urban sites comparison based on a two-sample t-test (t = 2.342; df = 33.568; p = 0.025). The δ15N of rural domestic herbivores is on average 0.8‰ higher than that of urban fauna; the effect size of this difference in nitrogen isotope values was moderate to large (Hedges' g = 0.7).”

- Line 435 = ??? I don't know what that means.

We have changed the line to:

 “At the same time, differences in the domestic herbivore δ15N values, not significant when compared with Kruskal–Wallis test between the groups of sites mentioned above (H = 6.052; df = 3; p = 0.109; Table S4), are significant at the level of the rural to urban sites comparison based on a two-sample t-test (t = 2.342; df = 33.568; p = 0.025). The δ15N of rural domestic herbivores is on average 0.8‰ higher than that of urban fauna; the effect size of this difference in nitrogen isotope values was moderate to large (Hedges' g = 0.7).“, which should hopefully be clearer.

- Line 437 = on average

Added “on average”.

I would test for effect size here, as t-tests are very susceptible to large sample sizes when it comes to isotope data and a difference of <1 permil is not a lot of for d15N values..

This may be a lot more convincing in a plot of the data, of course.

We have now added the effect size measures to the t-test results and changed the line accordingly:

“At the same time, differences in the domestic herbivore δ15N values, not significant when compared with Kruskal–Wallis test between the groups of sites mentioned above (H = 6.052; df = 3; p = 0.109; Table S4), are significant at the level of the rural to urban sites comparison based on a two-sample t-test (t = 2.342; df = 33.568; p = 0.025). The δ15N of rural domestic herbivores is on average 0.8‰ higher than that of urban fauna; the effect size of this difference in nitrogen isotope values was moderate to large (Hedges' g = 0.7). “

We have also added the description of this calculation to the “Statistical analysis” section of the “”Materials and methods”:

“To assess the magnitude of differences in the mean values for the independent samples t-test with varying sample sizes and standard deviations, we calculated the effect size index Hedges' g with the online software Social Science Statistics [92].”

The comparisons between sites are now plotted in Figure S1.

- Line 437-441 = Once again clear what you compare: presumably these are still domestic herbivores, as you would not just pool all domestic animals from one period regardless of species.

Thank you for pointing out this discrepancy; as we have checked, we indeed compared the pooled sample of domestic animals in this instead of the domestic herbivor sample as everywhere else. We have recalculated the statistics and changed the corresponding results and discussion as follows:

Results: “Comparing the set of domestic herbivore samples by period with a Kruskal–Wallis test shows that there are differences (H = 15.319; df = 2; p = 0.001) in δ13C values between the 11-13th centuries (n = 26) and both the 13-16th centuries (n = 17) and the 16-18th centuries (n = 6), with the mean of the High Medieval fauna being 1‰ higher than that of the Late Medieval period and 1.4‰ higher than that of the Early Modern period (Fig 3, Table S5). A Kruskal-Wallis test indicates no statistically significant differences in δ15N values between the time periods (H = 1.433; df = 2; p = 0.565; Table S6).”

Discussion: “A comparison of domestic omnivore (pig) δ13C and δ15N between sites and chronological periods indicates no differences, while the domestic herbivore sub-sample showed a slight, statistically significant increase in δ13C between the groups of sites dated to the 11-13th centuries AD and the other two periods. Variation in environment or human management practices (e.g. millet foddering) between the geographic locations of sites (Biruli and especially Mienka are located to the south-west of the cluster of Polack, Lučna and Čarscviady, and have higher δ13C values) or between the periods to which they date are among the possible explanations for this difference. “

It has not affected the interpretation much, as it only strengthens the difference between the 11-13th century sites’ cluster and the fact that the faunal pattern of the changes in carbon values over time is opposite to that of the human one.

Figure 3 is unintelligible as far too full: Use averages for animals through time (and delete individual datapoints). Add mean and standard deviation for human groups but keep individual data points. Remove the fish from this graph (you have it in others) and adjust the axes to focus on the human data.

We have now applied the recommended changes (see response regarding this figure to Reviewer 1 above) and have changed the caption accordingly:

“Fig 3. Plot of δ13C and δ15N values of human and animal samples (mean ± 1 std. dev.) with human and herbivore samples compared by period. Smaller triangles show the individual values of human samples.”

- Line 444 = Test statistic is H

Thank you, added the name of the statistic to results in all cases.

- Line 451-453 = Can you verify that this sample is, indeed, from a human? If you cannot, it would be much less confusing to note from the start that it has been excluded on that ground and not report it as part of the summary stats in the first place

This rib was initially identified as human by two analysts and after this comment the remains of it were double checked by one more analyst, and all believe it to be human. If possible we will verify it through DNA analysis when further expanding our research, but at the current moment it is out of the budget scope.

- Line 471 = again, effect size assessment needed. This is hardly a meaningful difference

We have now added the effect size measures to the t-test results and changed the lines in the results and the discussion accordingly:

“However, the difference in δ15N only remains significant in the sample dated to the 13-16th centuries AD (Student’s two-sample t-test, t = 2.766; df = 54; p = 0.008; Hedges' g = 0.8) when urban and rural groups are compared separately within each period.”

“This small difference is significant only in the 13-16th centuries; however, the effect size of this difference for the 13-16th centuries is large (Hedges' g = 0.8) and it is notable that at least the urban herbivores at this time have lower δ15N values.”

- Line 485 = coeval

Changed “contemporary” to “coeval”

- Line 485-490 = where are these data plotted? Readers may want to verify whether difference looks meaningful (this reader does!)

We have now added a new Figure S2 to the Supplementary Information where the groups of samples in question (elite, military, rural context) are shown alongside each other and with a regression line for the elite sub-sample. We have also added a reference to the new Fig S2 in this line.

- Line 504-510 = yet you said there were site-based differences before (line 433)? Clarify.

Here we speak of the omnivore sample (added “domestic omnivore” now to also address the previous concerns of the reviewer), while in line 433 we referred to the differences between domestic herbivores, which are discussed later within the same line: 

“A comparison of domestic omnivore (pig) δ13C and δ15N between sites and chronological periods indicates no statistically significant differences, while the domestic herbivore sub-sample showed a slight, statistically significant increase in δ13C between the groups of sites dated to the 11-13th centuries AD and the other two periods. Variation in environment or human management practices (e.g. millet foddering) between the geographic locations of sites (Biruli and especially Mienka are located to the south-west of the cluster of Polack, Lučna and Čarscviady, and have higher δ13C values) or between the periods to which they date are among the possible explanations for this difference.”

- Line 512-513 = But: You would expect cattle and ovicaprid remains from towns to belong to animals reared in the countryside, would you not?

We agree with the Reviewer 2 that most of the animals (and food products in general) may be expected to come to the town from the countryside, though it is also discussed that animals were kept in Polack, and its inhabitants had their own gardens, so were involved in husbandry and agriculture like the rural dwellers, especially at the earlier centuries’ of the city’s existence. Therefore it makes an interesting comparison to check for any difference between the rural and the urban animal sample that could be detectable through isotopes. To acknowledge this aspect, we have added a further line to this paragraph:

“Rural herbivores also have a higher δ15N, possibly due to obtaining more of their fodder from manured fields. Though we have previously discussed that city inhabitants increasingly obtained their food from rural neighbourhoods, agricultural activity was common in cities in the 10-16th centuries and city inhabitants also kept their own gardens and domestic animals [7,9,29,30,33,37,41], which may explain why there are differences between the animals from rural and urban contexts.”

- Line 518-519 = Consider other environmental/ anthropogenic explanations for high d15N than manuring

We have considered here several common reasons that could have led to nitrogen enrichment, including “extreme protein stress like starvation, or being fed a specific diet, or perhaps due to this individual coming from some distance away where local baseline δ15N values were higher due to manuring”. 

We agree that a more detailed consideration of this animal may bring up a broader variety of reasons, but consider such a discussion beyond the scope of this article’s topic. We still believe that manuring is one of the possible reasons for this animal’s enrichment in 15N - given the wet environment of Belarus and the surrounding geographic region, aridity is less likely.

- Line 539-540 = Verify that this sample is indeed demonstrably from a human

See the reply to the same comment above (Line 451-453).

- Line 568 = dietary variation between males and females were observed…

Thank you for this suggestion for improvement, we have changed the line as proposed: 

“No evidence for dietary variation between males and females was observed, either in the whole sample or within its subgroups based on urban or rural context or time period.”

- Line 579-581 = Clarify: consistent with forest feeding? You never really discussed this. Depends on the species in question though (hence importance of giving them)

We had not discussed the canopy effect, as no significant differences in C were observed between the wild and domestic herbivores, and the wild species (now listed in the text as per earlier comment) include both those from woodland and near woodland environment (e.g. red deer, bison, elk) and grassland or wetland (hare, beaver, wild bird). The wild sample is rather small and diverse thus can’t be directly connected to woodland or statistically analysed by species. Therefore, we discussed the difference in stable nitrogen isotope values as related to feeding on manured fields versus any environment not so affected by humans. To address the concern of the lack of relevant discussion, we have now added this considerations to the Discussion section:

“Although wild fauna do have lower δ13C values than domesticated fauna, this difference is not statistically significant. Furthermore, our small sample of wild fauna includes both species from woodland and near woodland environments (e.g. red deer, bison, elk) and grassland or wetland (e.g. hare, beaver, wild bird). As a result, there does not seem to be an obvious impact of the canopy effect in our study, though larger analyses in the future could explore this further.”

To avoid creating confusion, we have also deleted the “δ13C” from the sentence in question, as the difference in carbon isotope is not significant, while we primarily discuss the nitrogen δ15N in this context:

“A possible explanation could be that the nobility were consuming greater amounts of wild animal protein (which seems to overall have lower δ15N values in our study), which could offset increased animal protein use among elites and further highlight the significance of hunting as a sign of status in the region.”

- Line 591 = play

have someone edit this whole paper thoroughly at the end

Thank you, we have changed the line as suggested. Both of the native English speakers among co-authors reviewed the text, but some mistakes may still have been missed.

- Line 603 = what does that mean?

We have deleted this sentence while applying changes following the Reviewer 1’s comments to this paragraph.

- Line 669-674 = The apparent size of the effect is nevertheless surprising. I have never seen humans from this time period with such low carbon isotope values - and the classical studies on spatial variation (eg. van Klinken et al 1994) do not suggest that differences are so substantial.

I would include more detail on the environmental differences to explain.

We agree that this phenomenon is interesting and therefore consider it worth a separate investigation, which some of the authors intend to pursue in the future. At the moment we consider further study of this phenomenon out of the scope of this article, as it would indeed require greater environmental knowledge and investigation.

Also: Have any standards been re-calibrated between these datasets being produced? CH-7

No standards were re-calibrated between data sets.

- Line 1037 = ??? PRovide details

Thank you for pointing this out, we have fixed the reference to “Chesson L, Meier-Augenstein W, Berg G, Bataille C, Bartelink E, Richards M. Basic principles of stable isotope analysis in humanitarian forensic science. In: Forensic Science and Humanitarian Action. 2020. pp. 285–310. doi:10.1002/9781119482062.ch20“

- Line 1039-1040 = This should be referenced as a monograph, should it not?

Thank you for pointing this out, we have fixed the reference to “Scheuer L, Black S. Developmental Juvenile Osteology. London: Academic Press; 2000.“

---

## [Decision Letter · Decision Letter 1]

22 Sep 2022

Medieval and Early Modern diets in the Polack region of Belarus: A stable isotope perspective

PONE-D-22-08905R1

Dear Dr. Haponava,

We’re pleased to inform you that your manuscript has been judged scientifically suitable for publication and will be formally accepted for publication once it meets all outstanding technical requirements.

Kind regards,

Luca Bondioli, M.D.

Academic Editor

PLOS ONE

Additional Editor Comments (optional):

Reviewers' comments:

Reviewer's Responses to Questions

**Comments to the Author**

1. If the authors have adequately addressed your comments raised in a previous round of review and you feel that this manuscript is now acceptable for publication, you may indicate that here to bypass the “Comments to the Author” section, enter your conflict of interest statement in the “Confidential to Editor” section, and submit your "Accept" recommendation.

Reviewer #1: All comments have been addressed

2. Is the manuscript technically sound, and do the data support the conclusions?

Reviewer #1: (No Response)

3. Has the statistical analysis been performed appropriately and rigorously? 

Reviewer #1: (No Response)

4. Have the authors made all data underlying the findings in their manuscript fully available?

Reviewer #1: (No Response)

5. Is the manuscript presented in an intelligible fashion and written in standard English?

Reviewer #1: (No Response)

6. Review Comments to the Author

Reviewer #1: Haponava and colleagues have addressed all the comments highlighted exhaustively. As a result, the manuscript has significantly improved, and the figures look much clearer than in the previous version. It follows just a comment to be revised:

A) p.155 about their comment “To address the rightful concern of the Editor and Reviewer 1, we have replaced the narrow age ranges in the “age” column of Supplementary Table 1 with the age categories “adult” and “adolescent” to avoid providing unjustified information.”

Unfortunately, I cannot see the difference in Table S1.

7. PLOS authors have the option to publish the peer review history of their article (what does this mean?). If published, this will include your full peer review and any attached files.

Reviewer #1: No

---

## [Editor Report · Acceptance letter]

27 Sep 2022

PONE-D-22-08905R1 

Medieval and Early Modern diets in the Polack region of Belarus: A stable isotope perspective 

Dear Dr. Haponava:

I'm pleased to inform you that your manuscript has been deemed suitable for publication in PLOS ONE. Congratulations! Your manuscript is now with our production department. 

Kind regards, 

on behalf of

Dr. Luca Bondioli 

Academic Editor

PLOS ONE